# Image steganography techniques for resisting statistical steganalysis attacks: A systematic literature review

**Richard Apau** [ID]*, **Michael Asante, Frimpong Twum, James Ben Hayfron-Acquah, Kwame Ofosuhene Peasah**

Department of Computer Science, Kwame Nkrumah University of Science and Technology (KNUST), Kumasi, Ghana

* rich4u34@yahoo.com

## Abstract

Information hiding in images has gained popularity. As image steganography gains relevance, techniques for detecting hidden messages have emerged. Statistical steganalysis mechanisms detect the presence of hidden secret messages in images, rendering images a prime target for cyber-attacks. Also, studies examining image steganography techniques are limited. This paper aims to fill the existing gap in extant literature on image steganography schemes capable of resisting statistical steganalysis attacks, by providing a comprehensive systematic literature review. This will ensure image steganography researchers and data protection practitioners are updated on current trends in information security assurance mechanisms. The study sampled 125 articles from ACM Digital Library, IEEE Explore, Science Direct, and Wiley. Using PRISMA, articles were synthesized and analyzed using quantitative and qualitative methods. A comprehensive discussion on image steganography techniques in terms of their robustness against well-known universal statistical steganalysis attacks including Regular-Singular (RS) and Chi-Square ($X^2$) are provided. Trends in publication, techniques and methods, performance evaluation metrics, and security impacts were discussed. Extensive comparisons were drawn among existing techniques to evaluate their merits and limitations. It was observed that Generative Adversarial Networks dominate image steganography techniques and have become the preferred method by scholars within the domain. Artificial intelligence-powered algorithms including Machine Learning, Deep Learning, Convolutional Neural Networks, and Genetic Algorithms are recently dominating image steganography research as they enhance security. The implication is that previously preferred traditional techniques such as LSB algorithms are receiving less attention. Future Research may consider emerging technologies like blockchain technology, artificial neural networks, and biometric and facial recognition technologies to improve the robustness and security capabilities of image steganography applications.

**Data Availability Statement:** All relevant data are within the manuscript and its Supporting Information files.

**Funding:** The author(s) received no specific funding for this work.

**Competing interests:** The authors have declared that no competing interests exist

## 1. Introduction

Information technology has revolutionized many aspects of the human society. Presently, computing technologies have permeated our daily activities including shopping, banking, education, and communication [1]. These technologies have boosted productivity and automated many tasks. With the increased pervasive network connectivity and technology convergence, an enormous amount of information is produced, processed, stored, and shared every day [2]. For example, Facebook sees over 147, 000 pictures uploaded every 60 seconds [3]. Organizations rely heavily on information technologies for communication and information sharing [4]. Technological platforms such as email, videoconferencing, and social media apps are widely used by organizations to facilitate employee information sharing, meetings, and/or public product advertising.

While information sharing through computing technologies has its benefits, it is also susceptible to various threats such as cyber-attacks, data theft, and data breaches [5]. Numerous reports exist regarding data leakage, data loss, and unauthorized access to confidential information in digital communication [6,7]. Data breaches have affected many companies and organizations across different sectors, resulting in multimillion-dollar losses to cyber criminals [8]. Cybercrime Ventures [9] estimated annual cost of data breaches to reach 10.5 trillion United States dollars globally by 2025. Records totaling 4.5 billion were exposed by mid-2018 alone, whereas in 2019 identity records totaling 2.7 billion were exposed [10]. For example, the Thales 2022 data threat report revealed that 45% of companies in the United States experienced data breaches [11]. Additionally, in 2022, T-Mobile data breach pay-outs to customers and regulation fines cost the company 350 million dollars [12]. An Analysis by Nallainathan [13] projected a rise in cyber-attack trends in the next decade. As organizations suffer these occurrences, they incur significant financial and reputational losses [13]. According to Bouveret [14], more than 1 billion US dollars has been lost by financial institutions since 2010. Further, the operations of many institutions are threatened by these threats as cyber-attacks continue to grow more complex and sophisticated. Poor security measures are at the heart of many of these data breaches. Consequently, securing communication and information exchange has thus become paramount.

Given the rapid pace of data compromises and the potential threats to the security of individual and organizational data, steganography, which is an information-hiding technique, and cryptography, a data protection approach has gained notable attention in recent years. While cryptography ensures data confidentiality by altering the meaning of the message being transmitted, steganography conceals the existence and contents of secret information [15]. In other words, cryptographic techniques transform the message such that its original meaning is obscured from an unauthorized entity [16] and steganography covertly embeds the message within an innocent-looking cover (or media) [17]. Although cryptography is effective in securing communication channels, it is limited because the jumbled messages arouse suspicion in the minds of intruders, who potentially may destroy the message [18]. Hence, the intended recipient may not get access to the message. Also, a technique called cryptanalysis serves as a countermeasure against cryptography with the intended aim of revealing a secret message, thereby undermining the security, privacy, and secrecy of the message [19–26]. Steganography therefore provides another layer of security to enhance the protection of data against unauthorized access and use. Steganography is effective for ensuring confidentiality, integrity, and availability [27].

Steganographic applications are categorized into five types. These are image steganography, network protocol steganography, text steganography, video steganography, and audio steganography [1]. However, image steganography has gained the most popularity due to the degree

of redundancy associated with images [28]. As image steganography continues to gain relevance as an effective approach in the field of information security, techniques for detecting hidden messages have emerged. Specifically, steganalysis is a technique that aims at uncovering and extracting hidden messages from a cover (or media) that is gaining prominence in the domain [18]. Statistical steganalysis mechanisms such as RS attacks detect the presence of LSB-based hidden secret messages [29]. These mechanisms have exposed image steganography, rendering images a prime target for cyber-attacks. Given the rapid advancement and increasing sophistication of information technologies, steganalysis techniques are expected to grow more powerfully [15]. For the image steganography technique to be efficient, resistance against universal steganalysis attacks is paramount. Consequently, more robust image steganography techniques capable of withstanding statistical steganalysis attacks are urgently needed. A comprehensive understanding of image steganography techniques for resisting statistical steganalysis is required to safeguard information against detection, alteration, and modification and to guarantee data protection assurances and enhanced information security.

Yet existing studies that examine image steganography techniques are limited, and relevant review studies fail to provide detailed empirical-based discussions on issues related to image steganography techniques. In other words, existing studies have not adopted a standardized methodology for reviewing the selected publications [30–33]. For instance, Bhattacharyya and Banerjee [30], Febryan et al., [31], and Shehab and Alhaddad [34] all conducted review studies that employed steganography techniques to hide data in image, audio, and video but none of these studies adopted an empirical approach or standardized method for selecting the studies, potentially introducing errors, omissions, and biases that hinder informed decision-making.

This empirical systematic literature review aims to fill the existing gap in the literature and provides a comprehensive literature review on image steganography schemes proposed to resist statistical steganalysis attacks. Systematic literature reviews on image steganography techniques are limited, and the existing review studies do not provide an adequate and comprehensive understanding of the phenomena. This paper provides a holistic overview of the field's advancements, methodologies, challenges, and emerging trends in statistical steganalysis attacks. The major contribution of this paper is as follows:

- A systematic literature review of image steganography techniques capable of resisting steganalysis attacks is presented. Research articles from four reputable electronic databases comprising ACM Digital Library, IEEE Explore, Science Direct, and Wiley are selected.

- Comprehensive analysis using quantitative and qualitative methods and tools is conducted on the selected articles to develop patterns, trends, techniques, methods, and performance of existing image steganography applications using standard evaluation metrics. This is intended to help information security practitioners and data protection scholars to be abreast with existing data protection schemes and measures.

- Extensive comparisons are drawn among existing techniques to evaluate their merits and limitations as well as their robustness against statistical steganalysis attacks.

- Finally, based on the analysis and findings, future directions would be provided in the field of image steganography aimed at guiding researchers and scholars to set the direction on emerging technologies and approaches that could be adopted for future research to improve security within the image steganography domain.

The rest of the paper is structured as follows: Section 2 of the paper provides an overview of background literature on image steganography and statistical steganalysis attacks, as well as discussions on existing review works and their limitations. The review methodology using

PRISMA as demonstrated in **Fig 2** is presented in Section 3. In section 4, comprehensive results following the qualitative and quantitative analysis are elaborated including future scope and research directions, whereas section 5 discusses the results and presents implications for the study findings. Finally, section 6 provides key findings, conclusions, limitations, and recommendations for future research studies.

## Background literature

### 2.1 Image steganography

Information hiding in images has gained popularity in recent times [35]. Images have become important carriers to hide secret messages without changing the visual features and/or properties. As a result, images have become popular and widely used for steganography due to the degree of redundancy associated with them [36]. All image file formats are suitable for image steganography. File format types including TIFF, JPEG, PNG, GIF, and BMP are all appropriate to use. [37]. It is worth noting that each image file format has its advantages and disadvantages when employed for steganography purposes. Given that pixel values are utilized for image steganography, variations in pixel intensities between the original cover image and stego-images are sometimes experienced. The intensity variation is nonetheless subtle such that the undetectability and imperceptibility to the human visual system is achieved [38,39].

The commonality of images for steganography has subjected images to several targeted cyber-attacks including visual and statistical steganalysis attacks [40]. These attacks possess the ability to unearth concealed messages within images using steganalysis algorithms. Statistical steganalysis capabilities aimed at revealing hidden data in images include detection, extraction, disabling, and destruction of hidden data [41]. Tools and techniques used for such capabilities include lossy compression, denoising, image enhancement techniques, image approximation techniques, and geometrical modification [35]. These tools and techniques expose the vulnerabilities of image steganography on the digital landscape, rendering images a prime target of cybercriminal activities.

Image steganography uses three main traditional approaches (i.e., spatial domain, transform domain, and adaptive domain) to embed data [42]. The spatial domain approach entails the direct embedding of secret messages into image pixel values. This approach encompasses numerous techniques including the least significant bit (LSB) insertion algorithm [43–45], quantization-based methods [46], histogram-based methods [47], prediction error [48], modulo operations [49], and many other variations. Spatial domain methods have the advantages of high visual quality with minimal distortion effects, and high embedding payload capacity [38]. However, the spatial domain is less robust, making it susceptible to various forms of manipulation and attacks [38].

Given the challenges associated with spatial domain approaches, transform domain techniques emerged as a compelling alternative for secret data embedding [50]. The transform domain utilizes frequency sub-band coefficients to insert the secret message bits [51,52]. Although the data embedding and extraction processes are intricate compared to the spatial domain, this approach bolsters system security [50]. This embedding technique possesses the capability to withstand data manipulation approaches such as cropping, scaling, compression, and rotation. Some existing transform domain algorithms include Discrete Cosine Transform (DCT) [51], Discrete Fourier Transform (DFT) [53], Integer Wavelet Transform (IWT) [54], and Discrete Wavelet Transform (DWT) [55] among others. This method offers competitive advantages over spatial domain approaches by enhancing the robustness of the steganographic applications. However, both spatial and transform domain approaches have limitations [56], particularly regarding the susceptibility of the cover image to data manipulation and

modification. Notwithstanding these limitations, spatial domain methods such as LSB Insertion algorithm and Pixel Value Differencing (PVD) remain the most prevalent data embedding techniques for steganographic applications [57]. The spatial domain method alters the LSBs of the carrier image by directly replacing the LSBs of the original cover image with the secret message bits, while transform domain randomizes all the bits in the carrier image [58].

Considering the intricacies associated with spatial and transform domains, the adaptive domain method also known as the model-based method or masking has surfaced. This method employs dynamic techniques for pixel selection for data embedding and estimating an allowable number of bits that can be hidden within the carrier object [50]. Examples of this method include artificial intelligence, blockchain technology, machine learning, and genetic algorithms. Recent innovations have seen the implementation of biometric techniques and facial recognition technologies for image steganography, contributing to the security enhancement and robustness [59–63]. Adaptive techniques have a comparative advantage over spatial and transform domains due to their robustness and the ability to avoid detection by statistical steganalysis attacks. This method is also able to efficiently balance the tradeoffs between embedding capacity and security. The trade-off high embedding capacity on one side and security and robustness improvement on another side, remains a challenge in image steganography applications, for which constant innovations are required.

## 2.2 Statistical steganalysis attacks

Steganalysis techniques undermine the security capabilities of steganography, as they detect messages concealed in images to reveal the message and estimate the size/length. Given that image steganography has gained prominence for secret information hiding, image steganalysis emerges as a countermeasure. Image steganalysis exploits image processing techniques such as cropping, filtering, and blurring to detect, extract, disable, or destroy hidden information within cover objects [64]. Steganalysis algorithms are extant, some of which include pixel difference histogram (PDH) analysis, sample-pair analysis, RS analysis, and Chi-square ($X^2$) analysis [58] among others. RS steganalysis can detect LSB-based substitution stego-images, whereas Chi-square analysis which is based on a statistical distribution of binary values (0s and 1s) can determine if the image intensities follow random or distributed patterns. Statistical steganalysis process extracts the statistical characteristics of an image to accurately detect and estimate the exact size of hidden messages within a stego image [65]. By so doing, the hidden information is unveiled, and their length estimated. This breaches the confidentiality requirement of data transmission. All types of steganalysis possess the capability to identify, detect, and extract secret information hidden within a carrier object. For instance, PDH analysis can analyze and detect PVD-based image steganography. The analysis focuses on searching for the algorithm employed for the secret message concealment.

Chi-Square ($X^2$) statistical steganalysis was proposed by Westfeld and Pfitzmann [66] with the ability to detect sequentially embedded messages within an image. This approach, however, could not identify the presence of hidden messages based on random embedding. Notably, Provos [67] improved the technique proposed by Westfeld and Pfitzmann [61] to have the ability to detect and estimate both sequentially and randomly hidden messages. The sample-pair technique proposed by Dumitrescu et al., [68], is also another effective approach to detecting hidden messages based on LSB steganographic hiding process. Among the various types of statistical steganalysis, the RS attack developed by Fridrich et al. [69] is the most effective and well-known steganalysis technique which possess the capability to detect and reveal secret messages embedded within an image. RS steganalysis technique detects both sequential and random embedded secret messages. Statistical attack techniques adeptly differentiate stego-images

containing secret messages from cover images. This is done by mathematically investigating the relationship that exists between adjacent pixel groups and the pixel values of the stego-image, and the cover image [70]. Following the earlier work by Fridrich et al. [69], several steganalysis techniques with improved performance and detection capabilities have emerged [65–69,71–77]. The growing sophistication, complexity, and accuracy performance of steganalysis techniques have meant that a more secure image steganography scheme is required.

## 2.3 Previous/Related works

Empirical studies providing systematic review on image steganography techniques and methods aimed at resisting statistical steganalysis attacks are limited. Existing studies have failed to provide detailed empirical-based discussions on issues related to image steganography techniques and lacked a standardized methodology for reviewing the selected publications/articles. Ashwin et al., [78] conducted a review of image steganography techniques as well as steganalysis techniques capable of detecting secret information hidden in images. The study identified research trends, challenges, methods, and techniques for image steganography. Although Ashwin et al., [78] study provided early perspectives to scholars on existing techniques for resisting steganalysis attacks, the study was limited to only two embedding process approaches (i.e., spatial and transform). The study failed to provide broader insights into other notable techniques and algorithms dominating the field. The study also failed to adopt a standardized methodology for conducting the literature review. Subhedar and Mankar [79] focused on the issues and challenges of image steganography. The study provided key insights on image steganography performance evaluation metrics and explored various challenges that confront image steganography whose data embedding processes are based on spatial and transform domains. The study identified steganalysis techniques as key issues affecting the efficiency of steganography and provided future research direction. This study was however not systematic, as methods for selecting literature were not defined. The study also failed to discuss how existing techniques have performed against universal statistical steganalysis such as RDH and RS attacks.

Kadhim et al., [80] provided a review of image steganography techniques. The study discussed performance evaluation metrics as well as future research trends in the field of image steganography. The study provided key insights to researchers on the trends of digital image steganography but failed to provide a broader and comprehensive systematic review of key algorithms dominating the field. Standard methods were not applied in the selection of literature for the survey review. Mandal et al., [81] provided a review of digital image steganography tools available for embedding secret messages. The survey provided some image steganography techniques including adaptive and deep learning techniques and offered some key examples of some popular steganography tools. Comparison of the various tools were provided. Challenges of deep learning-based steganography were also enumerated. The study failed to adopt a standardized methodology for conducting the literature review and did not provide a comprehensive insight into all existing image steganography techniques/approaches. The study was limited to spatial and transform domain methods. Perhaps, the most comprehensive study and closely related to this paper is a systematic literature review conducted by Kaur et al., [50]. Kaur et al., [50] adopted standardized systematic literature review guidelines and selected 61 pieces of literature from four key databases comprising Web of Science, IEEE, Wiley, and ACM. The studies selected were published from 2011 to 2022. The results of the study show that extensive milestones for image steganography techniques have been achieved. Progress in all three data embedding processes (ie spatial, transform, and adaptive approaches) has seen notable improvement. The study further revealed that future research could focus on enhancing and striking an adequate balance between embedding capacity and robustness.

Other existing reviews focused on some specific domains within image steganography, further limiting the scope of the application of techniques for resisting statistical steganalysis. For example, Hussain et al., [82] provided a review on image steganography focusing on spatial domain techniques. The study highlighted some novel spatial domain techniques for image steganography including challenges and trends. Girdhar and Kumar [83] also provided a review of steganography techniques based on 3D images. Various 3D domain techniques including topological, geographical, and representation domains were discussed and compared in terms of payload capacity, resistance to attacks, and reversibility. Meng et al., [84] reviewed deep learning algorithm-based image steganography techniques. Various deep-learning algorithms were surveyed and discussed. Deep-learning algorithms used for coverless information hiding, steganalysis attacks, and watermarking were extensively presented and discussed. Qin et al., [85] comprehensively reviewed coverless image steganography techniques. The review provided a framework description of methods and techniques for coverless image steganography, highlighted recent developments in the area, and concluded that coverless image steganography provides resistance against steganalysis attacks.

Also, Puteaux et al., [86] focused their survey on reversible image steganography techniques. Techniques and methods compared included pixel value differencing or histogram shifting, re-echoing-based steganography, public key cryptography-based methods, prediction-based methods, and image partition-based techniques. Aslam et al., [87] conducted a review LSB based image steganography techniques. The review sampled 20 research studies published from 2016 to 2020. The 20 articles were further scaled down to 17 for the review. 20 data sets were identified for the evaluation of image steganography techniques. All the domain-specific studies reviewed [82–86] could not be conveniently classified as a systematic literature review except Aslam et al., [87]. The studies failed the threshold for systematic literature review when compared to the guidelines provided by Kitchenham and Charters [88]. The methods adopted for the study selection including inclusion and exclusion criteria, datasets, databases, data extraction methods, and queries were not detailed.

The above review works discussed may not be exhaustive for review research on image steganography techniques capable of resisting statistical steganalysis. However, the extensive literature search conducted in the most relevant scientific databases and libraries provided little evidence of a systematic literature review for image steganography techniques. The identified knowledge gap and other germane issues are the focus of this review. This research, therefore, seeks to conduct investigations into the literature on image steganography techniques capable of resisting statistical steganalysis attacks. By so doing, the review brings to the fore relevant studies on image steganography methods for resisting statistical steganalysis to bridge and/or expose the knowledge gap.

## 3. Review methodology

This research adopted a standardized methodology and procedure for the systematic literature review. The aim was to meet the objectives set out for the review. The study relied on PRISMA guidelines and procedures for conducting a systematic literature review. Many scholars have recently utilized PRSIMA for systematic literature review studies within the information technology landscape and was considered an effective and exhaustive framework for conducting systematic review studies [50,89–91].

### 3.1 Research approach

The PRISMA guidelines were chosen to ensure the review process is transparent, clear, and credible [92]. The processes involved in PRISMA include defining the systematic scoping

review, identifying potential studies through literature searches in relevant databases and electronic libraries using predefined keywords, abstract screening, selecting papers based on inclusion and exclusion criteria, article characterization, and mapping based on keywords and meta-analysis [93]. Based on the PRISMA guidelines, a data selection, extraction, and classification taxonomy were developed and implemented. The taxonomy defined review questions, literature search strategy, eligibility criteria for inclusion and exclusion, data analysis framework, and criteria for resolving opinion disparities among researchers.

## 3.2 Review research questions and protocol

Kitchenham and Charters [88] argued that review questions and review protocols are important components of the systematic literature review process as they reduce the researcher's biases and provide a critical framework to guide acceptable systematic reviews. Review questions are formulated during the initial stages of study planning to situate the study goals as the foundation upon which the study hinges [93]. This study adopted the Goal-Question-Metric approach suggested by Caldiera and Rombach [94] (See **Table 1** for the Goal-Metric Questions). This Goal-Question-Metric has previously been used by Lun et al., [95] and Wiafe et al., [96] as an efficient and effective approach for deriving systematic review objectives.

Statistical steganalysis attacks are growing at a tremendous pace. As such, techniques and methods for steganography that could withstand such attacks have become topical. Questions such as the most used image steganography techniques for resisting steganalysis attacks, the performance and security impact of image steganography techniques, and future scope and research direction for techniques within the image steganography domain remain critical and unanswered concerns that require addressing. These knowledge gaps need to be addressed. The review questions, the reason behind the questions, and the research approach to achieve the questions are listed in **Table 2**.

Following the formulation of the research questions and to further avoid biases in the literature search strategy, search terms and keywords, and study selection, the review protocol was separately developed by each of the members of the research team. The individual protocols were merged and further refined by the research team in a protocol development meeting. The merged protocol was refined, and the final protocol was adopted after an extensive review process and corrections. **Fig 1** provides a detailed diagrammatic representation of the final protocol adopted for the study demonstrating the main review processes followed.

## 3.3 Literature strategy

Brereton et al., [83] identified seven electronic databases as key for conducting exhaustive literature searches for studies within the information technology landscape and for software engineers specifically. These databases are IEEExplore, ACM Digital Library, Google Scholar, Citeseer Library, INSPEC, ScienceDirect, and EI Compendex. SCOPUS, Wiley Online, Web of Science (WOS), and Springer Link are also considered relevant electronic libraries [83].

**Table 1. Adopted Goal-Question-Metric [94].**

| The Purpose | The study analyses |
|---|---|
| The Issue | Trends in publication, application areas, techniques, security impacts, and future scope and research direction |
| The Object | Image steganography techniques for resisting statistical steganalysis attacks |
| The Viewpoint | From 2012 to 2023 |

**Table 2. Formulated review questions and motivation.**

| Item | Research Questions (RQ) | Rationale | Research Approach |
|------|-------------------------|-----------|-------------------|
| RQ1 | Q1. What have been the Trends in Publication of Image Steganography Applications? | This question aims to classify the reviewed studies including the publication outlets, country of origin of studies and yearly publication trends with the view of bridging the knowledge gap within the image steganography domain | Quantitative Approach |
| RQ2 | Q2. Which Methods and Techniques are Used in Image Steganography for Resisting Statistical Attacks? | This is aimed at identifying the various image steganography techniques and methods currently in use for resisting attacks. It would also provide analysis on the most dominant methods and classify them based on the embedding process. | Quantitative Approach |
| RQ3 | Q3. What are the Standard Performance Evaluation Metrics for Image Steganography Techniques | The motivation behind this question is to identify the current standard performance evaluation metrics that have been used to measure the performance of image steganography techniques. This is to provide researchers with the modern trends in existing image steganography technique evaluation | Qualitative Approach |
| RQ4 | Q4 What Security Impact Has the Techniques have on Image steganography for Resisting Statistical Attacks? | The rationale for posting this question is aimed at analysing and classifying the impact that the existing techniques and methods have had on resisting steganalysis attacks. This will allow researchers and data protection professionals to understand the advantages or strengths as well as the disadvantages or limitation of existing image steganography techniques and how best to bridge the gap | Qualitative Approach |
| RQ5 | Q5. What are the Future Scope and Research Direction for Image Steganography? | This question explores and identifies future possible research interest areas for scholars including new techniques and technologies that could be explores to enhance the attack resistant nature of image steganography. It also seeks to provide researchers with future aspirations on emerging areas of interest within the image steganography domain. | Qualitative Approach |

Before the actual search, a preliminary search was conducted on Google Scholar, Citeseer, and SCOPUS to identify the most appropriate databases, search terms, and search period. Based on the preliminary search, four (4) databases (i.e., IEEE, ACM Digital, ScienceDirect, and Wiley Online) were chosen. These electronic databases and libraries were chosen because they had the most relevant published studies on image steganography techniques. The keywords and search terms used for the database searches were made up of two categories. The categories were Steganography and related words (steganography, image, image steganography) and Steganalysis and related words (Steganalysis, statistical steganalysis, RS steganalysis). The search phrases were developed by combining words from both categories using the "AND" Boolean Operator. After several searches in databases by the researchers, five search terms were perceived as appropriate based on the results from the preliminary search. These terms were (i) "Steganography" and "Steganalysis" (ii) "Image Steganography" and "Steganalysis"

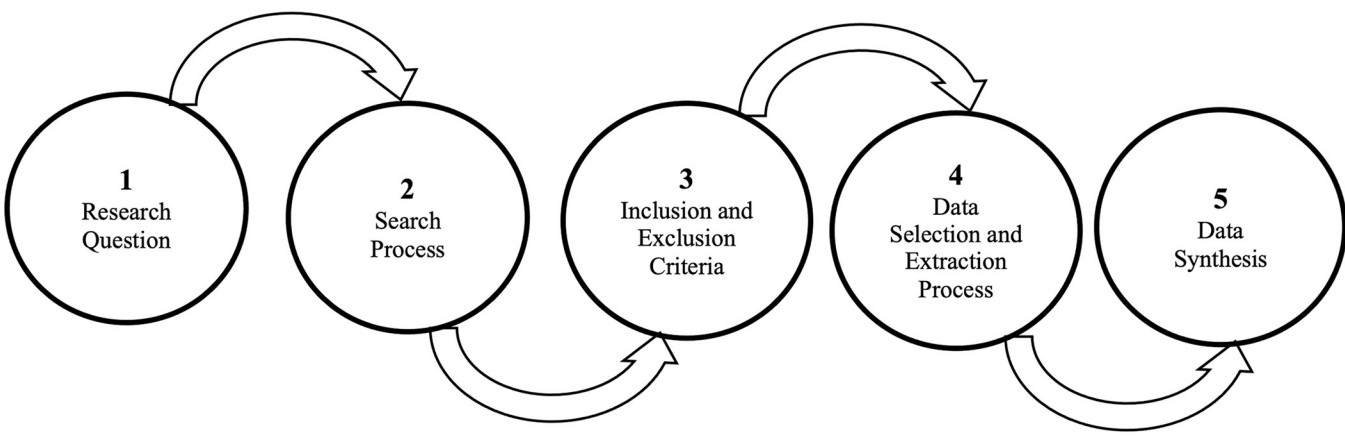

**Fig 1. Adopted review protocol for methodological analysis.**

(iii) "Steganography" and "RS Steganalysis" (iv) "Image Steganography" and "Statistical Steganalysis" and (v) "Image" and "Statistical Steganalysis". The search period was limited to 2012 to 2023 inclusive.

### 3.4 Eligibility criteria

For a publication to form part of this review, clear inclusion and exclusion criteria were defined. To be included, publications should have been written in English. Also, publications should have discussed image steganography and/or steganalysis attacks performance evaluation metrics. That is, publications whose titles related to image steganography and/or steganalysis attacks were included. Further, papers published from 2012 to 2023 were considered. Apart from these, only peer-reviewed publications were accepted. For the exclusion criteria, non-empirical studies were rejected. This suggests that point-of-view papers, review papers, and reports were excluded. Also, only peer-reviewed journal and conference papers were included. Book sections, chapters, posters, and thesis were excluded from the review. Moreover, publication abstracts that showed no relationship with the search terms were excluded. Publications whose content did not discuss how image steganographic techniques are employed to resist steganalysis attacks were removed. Lastly, publications ranked as low quality as agreed by the review team were excluded.

### 3.5 Study selection

Based on the search criteria, two (2) members of the review team performed independent searches using the identified search terms on all four (4) databases. For all searches, the search period was limited to 2012 to 2023 inclusive. The two (2) independent results were merged into one dataset. A total of 5146 publications were compiled. The dataset (n = 5146) was then screened to remove duplicates. After the duplicates were removed, 1379 publications remained. Next, the titles of the publications were scanned to determine their relatedness to the objectives of this review. For example, studies whose titles did not suggest any relation to image steganography techniques were removed. Next, the dataset was examined to maintain only journal and conference papers. Book sections, chapters, posters, and thesis were removed. Further, all non-empirical papers were discarded. This process reduced the total number of publications to 902. Reports were sought for retrieval and 13 reports were not retrieved. A total number of 889 records were maintained. After assessing the papers for eligibility, 736 papers were removed.

Two (2) members of the review team separately read the abstracts of the remaining publications (n = 153) to determine their relatedness to the search terms. The separate reports from the two (2) members were discussed by all members of the review team and merged. In cases of any disparities, a vote was conducted to resolve the issue. This activity further reduced the number of publications to 136. Lastly, two (2) other members of the review team read the content of the 136 publications to assess their quality. Their reports were also discussed and debated. Based on these discussions, 125 publications were retained as appropriate for review. **Fig 2** provides a detailed summary of the selection process for the identified publications. Thus, 125 papers remained as final papers included in the systematic literature review. Also, a summary of the number of papers selected from the various electronic databases and the search terms is shown in **Table 3**.

## 4. Results and analysis

### 4.1 Publication trends

The selected publications were analyzed to understand the publication trends. The information recorded for this analysis included the year of publication, publication outlet, publication

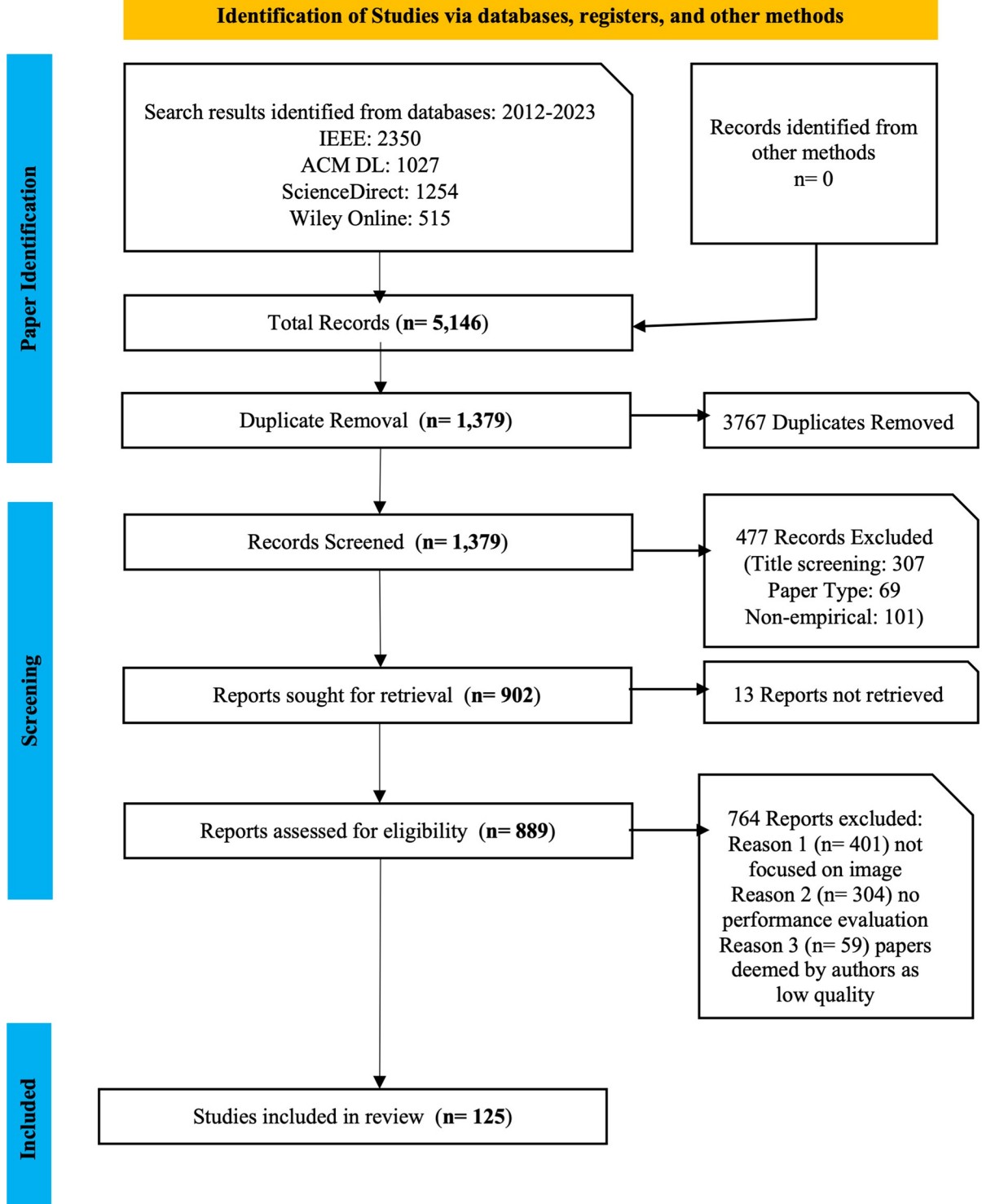

**Fig 2. PRISMA flow diagram for publication selection process.**

**Table 3. Detailed record of articles selected for the systematic literature review.**

| Electronic Database /Library | Shortlisting | Steganography AND Steganalysis | Image Steganography AND Steganalysis | Steganography AND RS Steganalysis | Image Steganography AND Statistical Steganalysis | Image AND Statistical Steganalysis | Total |
|---|---|---|---|---|---|---|---|
| ACM | Retrieved Articles | 454 | 135 | 122 | 302 | 14 | 1027 |
| | Selected | 3 | 2 | 1 | 4 | 1 | 11 |
| | Rejected | 451 | 133 | 121 | 298 | 13 | 1016 |
| IEEE | Retrieved Articles | 590 | 971 | 415 | 302 | 72 | 2350 |
| | Selected | 15 | 29 | 13 | 7 | 2 | 66 |
| | Rejected | 575 | 942 | 402 | 295 | 70 | 2284 |
| ScienceDirect | Retrieved Articles | 321 | 103 | 32 | 753 | 45 | 1254 |
| | Selected | 15 | 6 | 1 | 21 | 3 | 46 |
| | Rejected | 306 | 97 | 31 | 732 | 42 | 1208 |
| Wiley | Retrieved Articles | 125 | 190 | 116 | 51 | 33 | 515 |
| | Selected | 0 | 1 | 0 | 1 | 0 | 2 |
| | Rejected | 125 | 189 | 116 | 50 | 33 | 513 |

type, geographic origination of corresponding authors, and number of citations. The results show that publications on image steganography techniques for controlling statistical steganalysis attacks have increased considerably. For the year of publications, the results show fluctuations in the number of publications per year from 2012 to 2017 (**see Fig 3**). Since 2017, the number of publications per year increased tremendously. Articles published from 2018 to

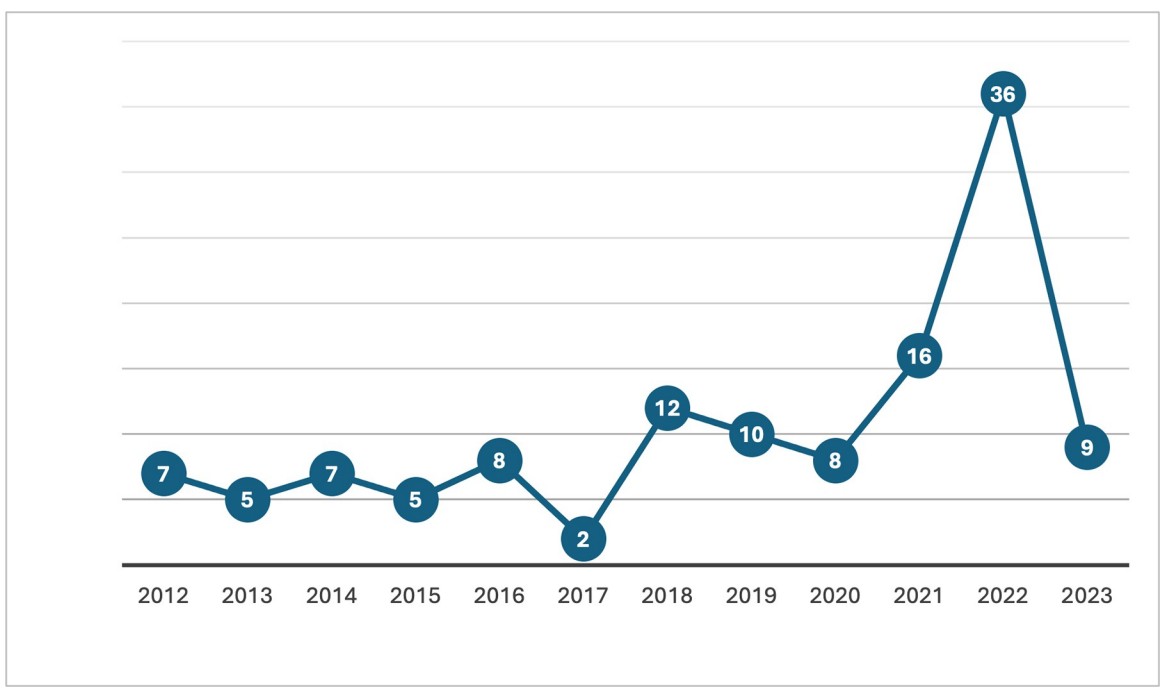

**Fig 3. Yearly publication trends of reviewed studies.**

2023 represented 73% of the total number of publications reviewed. This suggests a growing interest in image steganography studies for combatting steganalysis attacks. The analysis also shows an interesting result for the post-coronavirus Pandemic era (COVID-19), as approximately 49% of all articles were published from 2021 to 2023. This shows tremendous development of techniques against statistical attacks, following the numerous cyber-attacks, data breaches, and data compromises that were experienced during the peak of the COVID-19 lockdowns and global work-from-home phenomenon.

The results also indicated a skewed interest in publishing outlets. From the total of 125 papers reviewed, 66 (53%) were published with IEEE and 46 (37%) by ScienceDirect. **Fig 4** indicates the breakdown of the trend by publication outlet. Further, the analysis of the publication types revealed most of the reviewed publications were journals (57%) (n = 125).

Similarly, the results were geographically skewed. The affiliations of the corresponding authors at the time of publication were used to extract the geographic originations of the papers. The majority (86%) of the reviewed papers (n = 125) originated from Asia followed by Europe (8%). India (43 of 125) and China (37 of 125) recorded the highest number of publications respectively. **Fig 5** shows a summary of the geographical locations of all corresponding authors for the selected papers used for analysis.

The number of citations per paper at the time of this review was also analyzed. Majority (107 of 125) of the papers had 50 or lesser citations and only 8 had 100 or more. **S1 Appendix** shows the detailed list of the reviewed studies.

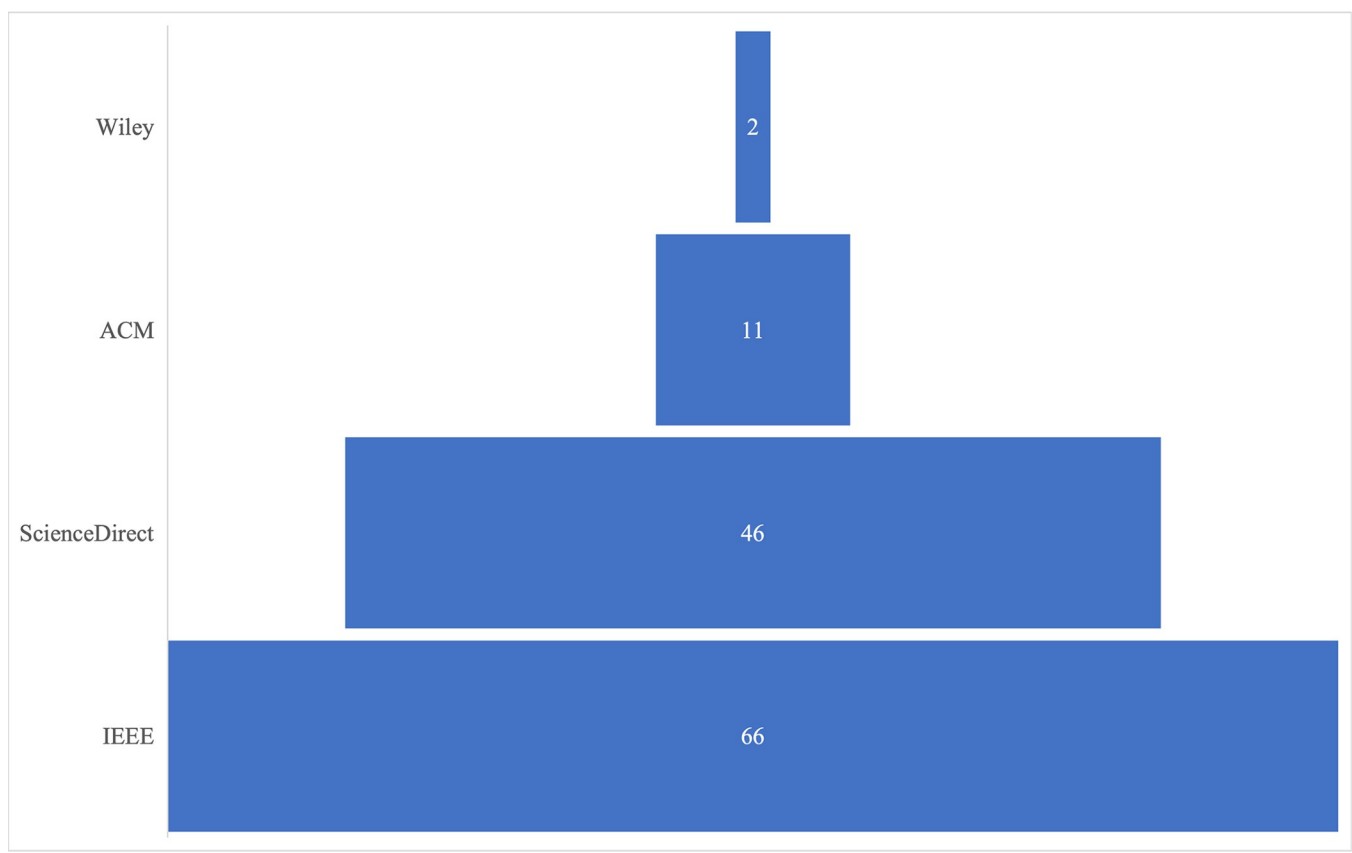

**Fig 4. Publication trend by publication outlet.**

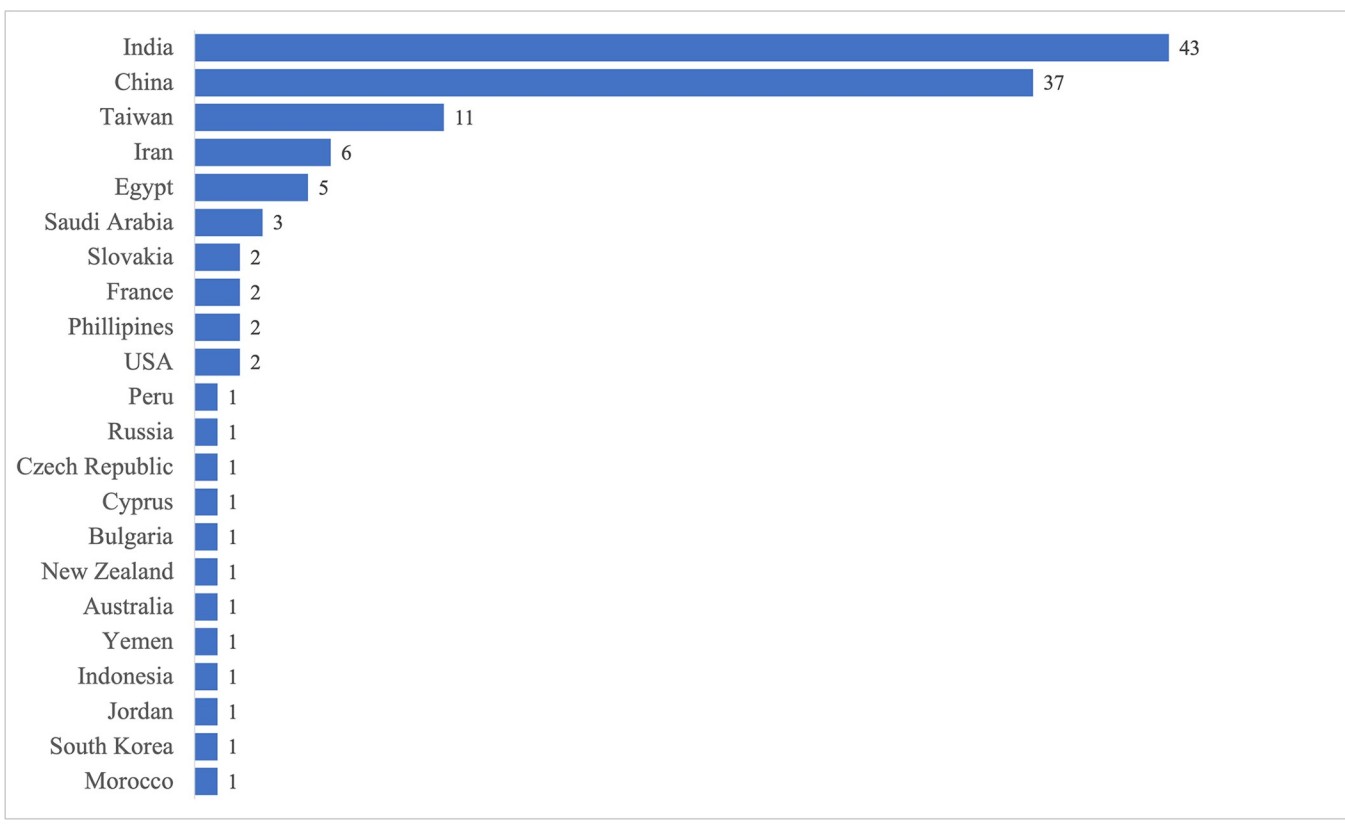

**Fig 5. Publication trend by geographic location.**

## 4.2 Image steganography techniques and methods

The review analyzed the methods and techniques that have been utilized in image steganography to resist statistical steganalysis attacks. Over 57 image steganography techniques and methods were identified. However, the techniques that have dominated image steganography studies are Modified LSB (M-LSB), LSB Matching (LSB-M), PVD, Genetic Algorithm (GA), GAN, CNN, DL Neural Networks, Hamiltonian Path (HP), Adaptive Edge Detection (AED), RDH, Residue Number System (RNS), DCT, IWT, among many others have been identified in literature as improving the imperceptibility of image steganography. Some of these methods have been implemented alone or sometimes with a combination of two of the methods enumerated. Others combined the methods with LSB and cryptographic protocols such as AES, RSA, and Elliptic Curve Cryptography (ECC) for encryption and decryption to enhance data security. As a result, many combinations of the above-mentioned techniques exist. The techniques and methods showed the capacity to enhance the visual quality of the carrier image and proved to be secure against statistical steganalysis attacks.

**Fig 6** shows that GAN (17) is the most adopted technique. This is followed by AED (14). A total of 20 studies implemented a version of LSB comprising M-LSB (4), LSB-M (10), and LSB plus others (6). GA, RDH, and PVD were each implemented in 9 studies. The techniques that were used by less than two publications were grouped as "Others". **Fig 6** gives details of the number of times other methods were utilized. Table 4 also gives a breakdown detail of the trend in publication year and techniques implemented. As already mentioned, the embedding process for image steganography techniques can be classified into three domains ie (i) Spatial

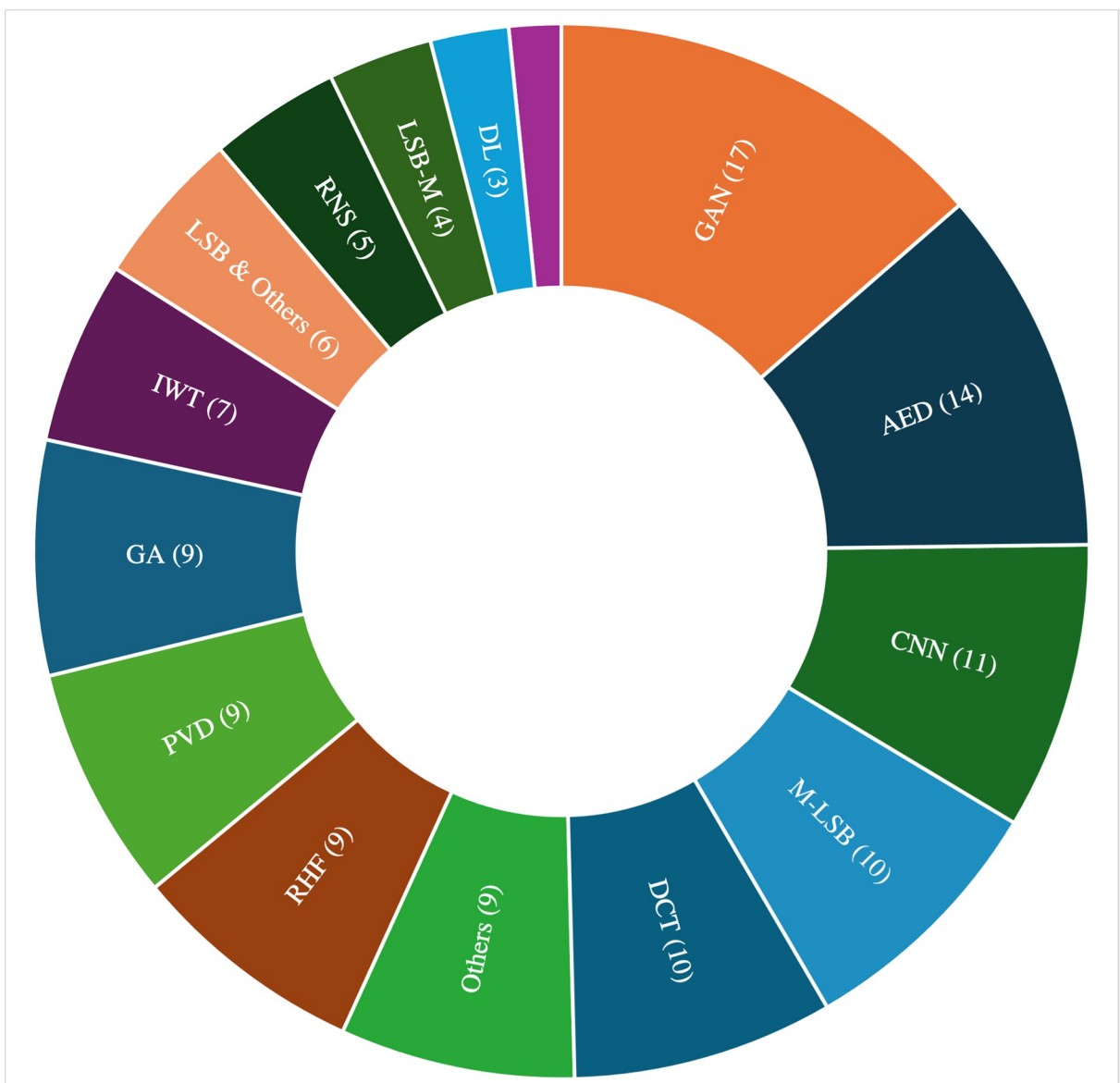

**Fig 6. Image steganography techniques and methods for resisting attacks.**

Domain-Based Techniques, (ii) Transform Domain-Based Techniques, and (iii) Adaptive Domain-Based Techniques. The review results reveal that spatial domain-based image steganography techniques have attracted more attention, as approximately 43% of all the reviewed papers utilized spatial domain for the secret data embedding process. This is followed by adaptive techniques, where 38% of reviewed papers employed such techniques. The rest of the studies used transform domain image steganography techniques (19%) (See **Table 4**). Further analysis of the review was conducted to understand the application of the image steganography techniques and the primary embedding domain employed for data hiding. This was necessary to observe the trend of specific techniques within each domain of application.

The results as presented in **Table 5** show that the spatial domain was the primary data embedding process for M-LSB, LSB-M, PVD, HP, LSB+Others, and AED. Also, almost all

**Table 4. Publication and image steganography embedding domains (2012 to 2023).**

| | Spatial Domain- Based Techniques | Transform Domain-Based Techniques | Adaptive Domain- Based Techniques |
|---|---|---|---|
| **2012** | 4 | 3 | |
| **2013** | 2 | 1 | 2 |
| **2014** | 4 | 2 | 1 |
| **2015** | 4 | | 1 |
| **2016** | 1 | 1 | 6 |
| **2017** | 2 | | |
| **2018** | 6 | 3 | 3 |
| **2019** | 4 | 2 | 4 |
| **2020** | 5 | | 3 |
| **2021** | 4 | 4 | 8 |
| **2022** | 15 | 7 | 14 |
| **2023** | 2 | 1 | 6 |
| **Total** | **53** | **24** | **48** |

papers whose techniques were based on GA, GAN, DL, and CNN utilized the adaptive domain as the primary process of data embedding. Similarly, for DCT and IWT techniques, the transform domain method was mainly used. For RNS and RDH techniques, the domain for data embedding process was varied, whereas most of the other studies employed spatial domain and adaptive domain for the embedding process. The implication is that the spatial domain has gained wide application in use for image steganography, perhaps due to its advantage of high embedding payload capacity. **Table 6** shows the trends in the year of publication versus image steganography techniques.

## 4.3 Performance evaluation metrics for image steganography techniques

The implementation of image steganography is aimed at achieving some key objectives. The key objective parameters are high embedding payload capacity, imperceptibility (visual quality

**Table 5. Embedding domains verses image steganography techniques.**

| | Spatial Domain- Based Techniques | Transform Domain-Based Techniques | Adaptive Domain- Based Techniques |
|---|---|---|---|
| **M-LSB** | 10 | | |
| **LSB-M** | 4 | | |
| **LSB+OTHERS** | 6 | | |
| **PVD** | 9 | | |
| **GA** | | | 9 |
| **DL** | | | 3 |
| **CNN** | | | 11 |
| **GAN** | | | 17 |
| **AED** | 14 | | |
| **RDH** | 2 | 5 | 2 |
| **DCT** | | 10 | |
| **IWT** | | 7 | |
| **RNS** | 1 | 2 | 2 |
| **HP** | 2 | | |
| **OTHERS** | 5 | | 4 |
| **Total** | **53** | **24** | **48** |

**Table 6. Image steganography techniques for resisting steganalysis attacks (2012 to 2023).**

| | M-LSB | LSB-M | LSB + Others | PVD | GA | DL | CNN | GAN | AED | RDH | DCT | IWT | RNS | HP | Others |
|---|---|---|---|---|---|---|---|---|---|---|---|---|---|---|---|
| 2012 | 1 | | | 2 | | | | | | 1 | 3 | | | | |
| 2013 | | | | | | | | | 1 | 1 | 1 | | | 1 | 1 |
| 2014 | | 1 | | | 1 | | | | 1 | | 1 | 1 | | | 2 |
| 2015 | 1 | | 1 | 1 | 1 | | | | 1 | | | | | | |
| 2016 | 1 | | | | 2 | | 2 | | 1 | 1 | | | | | 1 |
| 2017 | 1 | | | | | | | | | | | | 1 | | |
| 2018 | 2 | | | 1 | 1 | | 1 | 2 | 2 | | 1 | 1 | | 1 | |
| 2019 | | 1 | | | | 1 | 2 | 1 | 3 | | | 1 | 1 | | |
| 2020 | 2 | | | 1 | | | 1 | 1 | 2 | 1 | | | | | |
| 2021 | | | 3 | 1 | 1 | 1 | 1 | 4 | | | 2 | 2 | 1 | | |
| 2022 | 2 | 2 | 1 | 3 | 1 | 1 | 3 | 7 | 1 | 5 | 1 | 2 | 2 | | 5 |
| 2023 | | | 1 | | 2 | | 1 | 2 | 2 | | 1 | | | | |
| **Total** | **10** | **4** | **6** | **9** | **9** | **3** | **11** | **17** | **14** | **9** | **10** | **7** | **5** | **2** | **9** |

of resulting stego-image), robustness (distortion resistance), and security (un-detectability) among others. However, there is a trade-off between the performance evaluation parameters as most of the parameters result in opposite impacts with each other. For instance, techniques proposed to achieve high hiding capacity result in image distortion that ultimately reduces security and data protection. To achieve the objectives of image steganography techniques, various evaluation metrics are utilized. To measure imperceptibility, many studies have used Mean Square Error (MSE) [97], Peak-Signal-to-Noise-Ratio (PSNR) [98,99], Segmented Signal-to-Noise-Ratio (SNRseg) [100] and/or Signal-to-Noise-Ration (SNR) [101]. Also, Pearson Correlation Coefficient (NC) [102], Correlation Factor (r) [103], and Structural Similarity Index Measure (SSIM) [104,105] are used to measure the similarity between the cover image and the stego image to determine the image quality matrix. Bit Error Rate (BER) [106] is often used to measure the image distortion resistance, whereas Regular-Singular (RS) analysis [107,108] has proven effective in analyzing the detectability of the image steganography techniques against steganalysis attacks. Given that high embedding capacity is a key evaluation metric for image steganography techniques, Bits Per Pixel (BPP) is often used [109]. The dominant performance evaluation metrics for the reviewed papers, are PSNR, MSE, NC, SSIM, BPP, and RS analysis. The most used evaluation metrics are discussed below. However, the performance metrics used by each reviewed paper will be reported to ensure standardization and quality metrics comparison.

Imperceptibility is an important criterion in steganography [50]. Distortions between the original cover image (CI) and the resulting stego image (SI) must be relatively low to ensure higher imperceptibility of the image against attacks. Image Quality Measurement (IQM) is a mathematical approach to determining the quality of SI. When a secret message is embedded in the original selected CI, changes are noticed in the pixel values of the CI. Such changes affect the quality of the resulting SI. It is important to measure the changes in pixel values to ensure the SI is imperceptible. PSNR measures the distortion between CI and resulting SI. PSNR is determined using **Eq 1** written as [98]:

$$PSNR = 20 \cdot \log_{10}(MAX_I) - 10 \cdot \log_{10}(MSE) \tag{1}$$

$MAX_I$. represents maximum oixel value, whereas the **MSE** is Mean Square Error. The MSE measures of noticeable distortion between CI and SI. MSE is determined using Eq 2 [97]:

$$MSE = \frac{1}{mn} \sum_{i=0}^{m-1} \sum_{j=0}^{n-1} \left[ I(i,j) - K(i,j) \right]^2 \tag{2}$$

**M** and **N** represent the image height and width respectively. The lower the values obtained for MSE, the less distorted the difference between the CI and SI. Also, the higher the PSNR value, the higher the visual quality, thus higher imperceptibility.

Robustness of image the steganography technique proves that it is distortion resistant. To ensure that the technique is resistant to distortion, the similarity between the CI and SI is checked to determine whether the image has been distorted after embedding the secret message. SSIM is an important metric to check the structural similarity between the original CI and the resulting SI. The SSIM metric is calculated using **Eq 3**, and written as [103]:

$$SSIM(x,y) = \frac{(2\mu_x \mu_y + c_1)(2\sigma_{xy} + c_2)}{(\mu_x^2 + \mu_y^2 + c_1)(\sigma_x^2 + \sigma_y^2 + c_2)} \tag{3}$$

Where $c_1 = (k_1, L)^2$ and $c_2 = (k_2, L)^2$. $\mu x$ and $\mu y$ are the CI and SI mean intensity. The variances of x and y are represented $\eth^2_x$ and $\eth^2_y$ respectively, whereas $\eth_{xy}$ represents the covariance of x and y. the pixel values varying range is denoted by L, and the constant parameters are represented by $c_1$ and $c_2$. $k_1$ and $k_2$ values are always to taken to be 0.01 and 0.03 respectively. The NC also checks the distortion resistance between CI and SI. NC computes the degree correlation between the CI and SI, is determined using mathematical **Eq 4** as [102]:

$$NC = \frac{\sum^M \sum^N (X_{MN} - \bar{X})(Y_{NN} - \bar{Y})}{\sqrt{\sum^M \sum^N (X_{MN} - \bar{X})^2 \sum^M \sum^N (Y_{NN} - \bar{Y})^2}} \tag{4}$$

Where X is the CI, Y is the SI, $\dot{X}$ is the mean pixel intensity values for the CI, and $\bar{Y}$ is the mean pixel intensity values for the SI. Fundamentally, the image steganography technique aims to avoid statistical steganalysis attacks. As a result, one key parameter in the design is undetectability. Steganalysis attacks can have access to the data in transmission, thereby breaking the data confidentiality parameter. As already mentioned, Regular-Singular (RS) attacks are some of the well-known attacks. RS analysis is therefore performed to ensure the technique developed can resist statistical attacks. RS analysis is defined over three kinds of block flipping. The block flipping are positive flippings ($F_1$), negative flippings ($F_{-1}$), and Zero (0) flippings ($F_0$). $F_1$, $F_{-1}$, and $F_0$ become flipping functions and form what is termed a flipped group. The flipped group results from applying the flipping functions on each divided image block pixel value. **Eq 5** is for determining the flipped group function [70].

$$F(G) = (F_{M(1)(X1)}, F_{M(2)(X2)}, \ldots, F_{M(n)(Xn)}) \tag{5}$$

Where **M = M (1), M (2), . . ., M (n)** represents the flipped mask, and **M (i)** has values indicating either **1, 0,** or **-1**. G is regular if $f(G) < f(F(G))$ otherwise G is singular when $f(G) > f(F(G))$. The implementation requires first dividing the image into non-overlapping blocks and re-arranging each one of them into a vector G = (**X₁, X₂, X₃,** . . .**Xn**). The blocks are arranged in a zigzag scan order. The discrimination function of the pixel's correlation is measured using

**Eq 6** [**70**]:

$$f(x_1, x_2, \ldots, x_n) = \sum_{i=1}^{n-1} |x_i - x_{i+1}| \tag{6}$$

The pixel values are represented by *x* and *n* is used to represent the number of pixels Also, *f* represents partial correlation between the adjacent pixels. A smaller *f* value means a stronger correlation exists between adjacent pixel values. Payload capacity is an important measure for image steganography techniques. An algorithm for image steganography should be able to embed maximum secret messages without noticeable distortion. The overall effect, embedding the maximum payload capacity within the pixel values of the selected CI must be possible without distorting the visual quality of the resulting SI. Basically, the number of secret bits that have been hidden in the CI is the embedding payload capacity, which is calculated using BPP as shown in **Eq 7** and written as [**108**]:

$$bpp = \frac{Embedding\ Capacity}{M \times N} \tag{7}$$

Where M and N are the CI cardinality, and embedding capacity (EC) which refers to the number of secret bits that can be embedded within total CI pixel values is determined using **Eq 8** [**70**]:

$$Embedding\ Capacity\ (EC) = \frac{Number\ of\ Bits\ Used\ to\ Hide\ Data}{Total\ Number\ of\ Bits\ in\ Image} \times 100\%. \tag{8}$$

## 4.4 Performance metrics analysis

The performance evaluation metrics for all 125 reviewed papers are provided. The analysis covers the techniques employed, strengths, limitations, and results obtained in each reviewed paper. The problems or issues often discussed in image steganography research are diverse. Concerns such as the tradeoffs between embedding capacity and security, statistical attacks against image steganography systems, stego image distortion, low embedding capacity, and low visual image quality of stego images remain some key challenges and issues that are generally raised and discussed within the image steganography domain. As a result, most techniques are proposed to address these challenges. The analysis also covers the issues and problems discussed by the various articles that warranted the proposed techniques and methods. The reviewed papers are grouped according to the primary embedding process adopted. **Table 7** covers papers based on Spatial Domain-Based Techniques, **Table 8** covers papers based on Transform Domain-Based Techniques, and **Table 9** is based on Adaptive Domain-Based Techniques. The evaluation metric indicated in each reviewed paper is reported.

In order to compare the superiority of each of the methods mentioned in Tables **7–9** over other methods listed and to demonstrate the efficiency of each method through the approved standards (ie Payload Capacity, measured in Bit Per Pixel (BPP) and Imperceptibility using Peak Signal to Noise Ratio (PSNR) and measured in decibel (dB)), a graphical representation is provided. **See Fig 7.**

## 4.5 Security analysis of image steganography techniques

The security impact analysis examines the various identified techniques using some key parameters. **Section 4.4** has already provided a detailed review of all the 125 publications

**Table 7. Spatial domain-based image steganography techniques.**

| Reviewed Study (RS) | Year | Problem/Issue | Technique/ Method | Strength | Limitation | Evaluation Metric Results |
|---|---|---|---|---|---|---|
| RS43 | 2012 | Low visual quality | PVD, HVS and diamond Encoding (DE) | Improvement in visual image quality | The payload capacity is low | BPP = 1.000 PSNR = 37.66 |
| RS44 | 2012 | Trade-off between Security and Capacity | PVD | Successful secret image imperceptibility and high quality stego image | Payload estimation not offered | PSNR = 41.58 RS = 2.4% |
| RS66 | 2012 | Statistical Steganalysis Attacks | RDH and LSB | Capable of resisting both RS and Chi-Square attacks | Embedding capacity relatively low | Capacity = 90% PSNR = 50.51 RS = 6% |
| RS101 | 2012 | Statistical Steganalysis Attacks | M-LSB | High visual stego image quality | Cannot withstand complex RS steganalysis | Capacity = 497,849 PSNR = 31.69 |
| RS42 | 2013 | Low visual quality | HP and LSB | The technique produces minimum distortion on stego-image | Low embedding payload capacity | BPP = 1.000 PSNR = 52.52 MSE = 0.3640 |
| RS52 | 2013 | Low visual quality | AED and LSB-M | Robust against some known steganlaysis attacks | Low embedding capacity detected | Capacity = 10% RS = 1.5% |
| RS53 | 2014 | Low embedding capacity | AED and LSB | Good stego image quality | Noticeable image distortion with high payload | Capacity bits = 12929 PSNR = 40.79 |
| RS54 | 2014 | Stego Image Distortion | AED and LSB | Provided better security and minimised distortion | Very low payload capacity | BPP = 0.5000 RS = 0.17 |
| RS97 | 2014 | Trade-off between Security and Capacity | LSB-M | High quality visual image quality | Performance metrics extremely low below threshold | 0.2031 PSNR = 11.96 |
| RS45 | 2015 | Statistical Steganalysis Attacks | PVD and Patched Reference Table (PRT) | Difficult to detect by RS schemes | Noticeable distortion with high embedding rate | RS = 0.600 BPP = 0.800 |
| RS55 | 2015 | Trade-off between Security and Capacity | AED, LSB, Chaotic, and GA | Adequate balance between payload capacity and security | Realtime efficiency of algorithm is slow | BPP = 4.000 PSNR = 40.95 MSE0.3421 NC = 0.9048 SSIM = 0.9887 |
| RS102 | 2015 | Trade-off between Security and Capacity | M-LSB | High visual quality and better payload capacity | Algorithm execution time is high | Capacity = 262000 PSNR = 56.44 RS = 0.4345 |
| RS112 | 2015 | Statistical Steganalysis Attacks | LSB and Adaptive Key Technique | Ability to withstand steganalysis attacks and good embedding capacity | High values for computational complexity | BPP = 3.000 PSNR = 64.15 MSE = 0.2500 |
| RS56 | 2016 | Stego Image Distortion | AED, LSB and Symmetric Encryption | Produced imperceptible SI with minimal embedding distortion | High computational complexity | BPP = 3.000 MSE = 0.594 PSNR = 50.39 |
| RS103 | 2016 | Statistical Steganalysis Attacks | M-LSB and RSA | Very high SI quality and high imperceptibility | Payload is very low | PSNR = 74.02 |
| RS75 | 2017 | Statistical Steganalysis Attacks | RNS, Encryption and LSB | Robustness against statistical steganalysis attacks | Noticeable distortion with increased payload | Capacity bit = 131072 PSNR = 51.93 MSE = 0.4169 RS = 0.350 |
| RS104 | 2017 | Statistical Steganalysis Attacks | M-LSB and Contrast Stretching | Robust against RS attacks | Payload capacity is relatively low | Capacity = 30% RS = 0.0564 PSNR = 54.08 MSE = 0.0374 |
| RS41 | 2018 | Trade-off between Security and Capacity | HP and LSB | Achieved increased payload and high imperceptibility | Some complex known RS attacks can detect secret message | BPP = 3.000 PSNR = 39.39 NC = 0.9991 SSIM = 0.9870 |

(*Continued*)

**Table 7.** (Continued)

| Reviewed Study (RS) | Year | Problem/Issue | Technique/ Method | Strength | Limitation | Evaluation Metric Results |
|---|---|---|---|---|---|---|
| **RS46** | 2018 | Trade-off between Security and Capacity | PVD, LSB and AES | Robustness against attacks | Improvement of algorithm efficiency required | BPP = 4.000<br>PSNR = 36.38<br>SSIM = 0.9403<br>NC = 0.1465<br>RS = 0.35 |
| **RS57** | 2018 | Trade-off between Security and Capacity | AED, LSB and dilation operator | Improved embedding capacity with high imperceptibility and robustness | Low embedding capacity | BPP = 1.236<br>PSNR = 43.62<br>MSE = 2.824<br>SSIM = 0.9980 |
| **RS58** | 2018 | Stego Image Distortion | AED and LSB | Robustness and high visual stego image quality | Low embedding capacity | BPP = 0.300<br>PSNR = 57.33<br>RS = 0.0350 |
| **RS105** | 2018 | Low visual quality | M-LSB and Chaotic map | The application proved immune against visual degradation | The capacity is low | BPP = 0.900<br>PSNR = 44.09<br>SSIM = 0.9700 |
| **RS106** | 2018 | Statistical Steganalysis Attacks | M-LSB | Stego image have low probability of detection | Distortion noticeable with increased capacity | PSNR = 48.24<br>SSIM = 0.9935<br>RS = 0.4000 |
| **RS59** | 2019 | Statistical Steganalysis Attacks | AED, PVD and LSB | Resists various known steganalysis attacks and provide better visual quality | High estimated embedding time | Capacity bit = 105432<br>PSNR = 35.68 |
| **RS60** | 2019 | Low visual quality | AED and LSB | High imperceptibility and SI visual quality | The embedding time estimation is longer comparatively | Capacity bit = 183500<br>PSNR = 48.59<br>MSE = 0.8990<br>SSIM = 0.9982<br>NC = 0.1763 |
| **RS62** | 2019 | Low visual quality, Stego Image Distortion, and Statistical Steganalysis Attacks | AED | Stronger statistical security and better image visual quality | Low embedding payload | Capacity bit = 1000 |
| **RS98** | 2019 | Low visual quality, and Statistical Steganalysis Attacks | LSB-M and Image Enlargement | High capacity with preserved image quality | Time complexity is high | BPP = 4.000<br>PSNR = 49.40 |
| **RS47** | 2020 | Statistical Steganalysis Attacks | PVD, LSB and DE | Better image quality and robust against attacks | Embedding capacity results not presented | PSNR = 47.99<br>SSIM = 0.9883 |
| **RS49** | 2020 | Statistical Steganalysis Attacks | PVD, LSB and DL | High accuracy estimation rate | Distortion noticed with increased payload | BPP = 2.000 |
| **RS63** | 2020 | Trade-off between Security and Capacity | AED | The average execution time is very efficient | Embedding capacity is relatively low | BPP = 0.6500<br>PSNR = 48.61<br>MSE = 1.256<br>SSIM = 0.9986 |
| **RS69** | 2020 | Low visual quality | RDH, IWT and AES | Accurate reconstruction of reference image | Higher time complexity | Capacity = 100%<br>PSNR = 31.99<br>SSIM = 0.9323<br>RS = 0.9843 |
| **RS107** | 2020 | Trade-off between Security and Capacity | M-LSB and Pseudo Random Number Generator (PRNG) | Robustness against statistical steganalysis and increased capacity | The time complexity for the algorithm is high | BPP = 3.000<br>PSNR = 89.03<br>MSE = 0.0001 |
| **RS108** | 2020 | Statistical Steganalysis Attacks | M-LSB and PRNG | High imperceptibility and robustness | Embedding capacity not discussed | PSNR = 83.27<br>MSE = 0.0003<br>SSIM = 0.9999 |
| **RS48** | 2021 | Trade-off between Security and Capacity | PVD and LSB | Super high embedding rate capacity | Imperceptibility performance below threshold | BPP = 8.88<br>PSNR = 25<br>SSIM = 0.9999<br>NC = 0.8710RP |

(*Continued*)

**Table 7.** (Continued)

| Reviewed Study (RS) | Year | Problem/Issue | Technique/ Method | Strength | Limitation | Evaluation Metric Results |
|---|---|---|---|---|---|---|
| **RS50** | 2021 | Trade-off between Security and Capacity | PVD, IWT and LSB | Withstand some known steganalysis tools | Low stego visual image quality | BPP = 2.2800<br>PSNR = 33.83<br>SSIM = 0.9820<br>NC = 0.9970<br>RS = 0.1020 |
| **RS113** | 2021 | Statistical Steganalysis Attacks | LSB and AES | Enhanced security for secure data transmission | Performance metrics not discussed | N/A |
| **RS114** | 2021 | Statistical Steganalysis Attacks | LSB, AES, and Pixel Locator Sequence | Resistance to attacks and highly robust | The technique is not space-efficient | PSNR = 48.35<br>MSE = 0.9518<br>RS = 0.0275 |
| **RS115** | 2021 | Low visual quality, and Statistical Steganalysis Attacks, low embedding capacity | LSB, Random Number Generator and Range Technique | High imperceptibility and better embedding payload capacity | Time complexity for the algorithm is high | BPP = 2.9529<br>PNSR = 49.56<br>MSE = 0.0564<br>NC = 0.8256 |
| **RS65** | 2022 | Low visual quality, and Stego Image Distortion | AED and LSB | High capacity for hiding data | Image distortion and susceptible to RS attacks | Capacity bits = 5000<br>PSNR = 46.89 |
| **RS51** | 2022 | Low visual quality, Statistical Steganalysis Attacks | PVD and LSB | Resistance to known RS steganalysis attacks | Imperceptibility and visual quality image improvement required | BPP = 3.180<br>PSNR = 39.09<br>MSE = 0.4562<br>SSIM = 0.9986 |
| **RS71** | 2022 | Low visual quality, Statistical Steganalysis Attacks | RDH | Resist histogram and RS steganalysis attacks | Low embedding payload capacity | BPP = 1.43<br>PSNR = 43.13 |
| **RS72** | 2022 | Low embedding capacity | RDH and Encryption | High embedding capacity and robustness against attacks | Higher time complexity | BPP = 3.83<br>NC = 0.9822 |
| **RS99** | 2022 | Statistical Steganalysis Attacks | LSB-M and RDH | Better image quality | Low hiding capacity | BPP = 1.000<br>PSNR = 51.14<br>SSIM = 0.9983<br>RS = 0.543 |
| **RS100** | 2022 | Low embedding capacity, Statistical Steganalysis Attacks | LSB-M, RDH and PVD | Robust against some known statistical steganalysis | The embedding capacity is relatively low | BPP = 1.000<br>PNSR = 51.16<br>SSIM = 0.9942<br>RS = 0.3562 |
| **RS109** | 2022 | Statistical Steganalysis Attacks | M-LSB | Showed capacity to resist steganalysis | Performance evaluation metrics not discussed | PSNR mentioned but record not stated |
| **RS116** | 2022 | Statistical Steganalysis Attacks | LSB and DWT | Resistance to RS attacks and provided enhanced security | High time complexity and computational time | PSNR = 40.09<br>MSE = 0.2322<br>SSIM = 0.9988<br>RS = 0.2500 |
| **RS121** | 2022 | Stego image distortion | Digital Still Images | Provided higher resistance to detection | Low embedding capacity | BPP = 0.2900<br>PSNR = 45.05<br>NC = 0.9997 |
| **RS122** | 2022 | Statistical Steganalysis Attacks | Generic Steganography Algorithm (GSA) | Robust against steganalysis | Higher Computational Complexity | BPP = 3.100<br>PSNR = 69.45 |
| **RS123** | 2022 | Statistical Steganalysis Attacks | Uniform Payload Distribution (UPD) | Provides better distribution to better security | Embedding capacity is relatively low | BPP = 0.500<br>RS = 1.3151 |
| **RS124** | 2022 | Stego image distortion | Chaotic Encrypted Dual Radial Harmonic Fourier Moments | High robustness against attacks | Embedding rate not discussed | PSNR = 30.30<br>MSE = 0.4432<br>SSIM = 0.9776 |
| **RS125** | 2022 | Statistical Steganalysis Attacks | Intra-block Modification Optimisation (IbMO) | Improves security performance of image steganography | Time complexity is extremely high | BPP = 0.5000<br>PSNR = 40.12 |

(*Continued*)

**Table 7.** (Continued)

| Reviewed Study (RS) | Year | Problem/Issue | Technique/ Method | Strength | Limitation | Evaluation Metric Results |
|---|---|---|---|---|---|---|
| **RS61** | 2023 | Low visual quality, Statistical Steganalysis Attacks | RDH and Fuzzy Edge Detection | Robust against universal well-known attacks | High embedding capacity | BPP = 2.000 PSNR = 51.68 SSIM = 0.9931 RS = 0.4500 |
| **RS64** | 2023 | Statistical Steganalysis Attacks | Hybrid Edge Detection | Better robustness and high security | Low embedding capacity | PSNR = 57 SSIM = 0.9999 |
| **RS110** | 2023 | Statistical Steganalysis Attacks | Adaptive Error Correction | Robustness against Lossy JPEG compression | Performance evaluation metrics not discussed | BPP = 1.15 RS = 0.345 |
| **RS111** | 2023 | Statistical Steganalysis Attacks | LSB, AES, and Blowfish | Robustness against statistical attack | Low embedding capacity | PSNR = 85.64 MSE = 0.0001 |
| **RS118** | 2023 | Statistical Steganalysis Attacks | Guassian Edge Detection | Relatively high visual quality | Improved payload | BPP = 3.1270 PSNR = 36.4478 MSE = 0.7891 SSIM = 0.9593 |
| **RS119** | 2023 | Stego image distortion | LSB, Huffman Code, Encryption (MLE) | Adequate balance between security and embedding capacity | High computational complexity | PSNR = 83.99 MSE = 0.05 SSIM = 0.9999 |

retained for this study, which are presented along with their strengths and limitations. However, some other key indicators are relevant to determine how the various existing techniques can provide robustness and resistance against attacks and their overall security. This will also enable comparison among the reviewed papers using common standard metrics and parameters. The indicators assessed in this section include the image dataset employed for the experiment, type of data embedding process, data embedding style, secret image type, real-time implementation of a proposed algorithm or technique, application of cryptography protocol (encryption), data compression, values obtained for the PSNR, robustness against steganalysis attacks and the overall security of each technique.

Table 10 provides a detailed comparison of the various existing techniques reviewed which used grayscale images for the experiment whereas Table 11 provides a detailed comparison of the various existing techniques reviewed which used color images. The reviewed articles show that four benchmark datasets consisting of BOSS base, USC-SIPI, Seam Carving Original Q75, and 24 KODAK image Databases have widely been used. These databases contained specific images. The specific image dataset used by each reviewed article is reported. The data-hiding process is divided into spatial, transform, and adaptive domains. The data embedding style is divided into random and sequential. For secret image type, the categorizations are color or grayscale. Yes or no is used to represent whether the respective technique implemented the algorithm in real-time, whether encryption was applied to the secret data, and whether the secret data was compressed. Robustness against steganalysis attacks is divided into high, medium, and low. The specific parameters considered for the robustness are embedding process and style, secret image type, and encryption. Techniques that fully satisfy the evaluation criteria of the researchers considering the key parameters are rated high, those that partially satisfy are rated medium and those that least satisfy are rated low. Security of the reviewed articles is divided into good, average, and low. The overall security is evaluated by taking into consideration all the parameters previously discussed, most importantly PSNR values, Encryption, Real-time implementation, Compression, and embedding process. Other parameters discussed in section 4 (4.4) were also taken into consideration. The techniques that satisfy the maximum parameters as determined by the researchers are rated good. Those that satisfy the parameters partially are rated average, whereas those that least satisfy the key parameters are rated low. To

**Table 8. Transform domain-based image steganography techniques.**

| Reviewed Study (RP) | Year | Problem/Issue | Technique/ Method | Strength | Limitation | Evaluation Metric Results |
|---|---|---|---|---|---|---|
| **RS80** | 2012 | Statistical Steganalysis Attacks | DCT and IWT | High visual quality of SI and robustness against attacks | Embedding capacity not discussed | PSNR = 58.95<br>SSIM = 0.9999<br>RS = 4.20 |
| **RS81** | 2012 | Statistical Steganalysis Attacks | DWT | Improve security and distortion resistant | Embedding capacity not discussed | PSNR = 81.33 |
| **RS82** | 2012 | Statistical Steganalysis Attacks | DCT and AES | Increased security level for the steganography system | Embedding capacity not discussed | PSNR = 36.68<br>NC = 0.3906<br>SSIM = 0.5502 |
| **RS83** | 2013 | Statistical Steganalysis Attacks | DCT and LSB | Robust against low-pass filtering attacks | Embedding capacity not discussed | Uses Bit Error Rate (BER) |
| **RS84** | 2014 | Stego image distortion | DCT | Robustness against histogram analysis attack | Very low embedding rate | BPP = 0.100<br>PSNR = 43.97<br>RS = 0.143 |
| **RS90** | 2014 | Statistical Steganalysis Attacks | IWT | Robustness against attacks and high imperceptibility | Embedding duration is comparatively higher | Capacity = 95%<br>PSNR = 35.06<br>SSIM = 0.8723 |
| **RS68** | 2016 | Statistical Steganalysis Attacks | RDH | Improved security when compared to other methods | Time execution rate is low and embedding capacity is limited | BPP = 0.700<br>NC = 0.6239<br>PSNR = 47.64 |
| **RS85** | 2018 | Trade-off between Security and Capacity | DCT | Maintains minimum detectability against blind steganalysis attacks | Embedding capacity increased by 16.7% | PSNR = 53.38<br>MSE = 2.927 |
| **RS86** | 2018 | Statistical Steganalysis Attacks | DCT | Better robustness against common image processing attacks | The embedding rate is low | BPP = 0.7000 |
| **RS91** | 2018 | Trade-off between Security and Capacity | IWT and LSB | Better imperceptibility and higher embedding capacity | High computational complexity | BPP = 3.3438<br>PSNR = 32.4385<br>RS = 0.3600 |
| **RS76** | 2019 | Stego Image distortion | RNS | High visual quality for stego image | Image distortion with higher payload | BPP = 0.500 |
| **RS92** | 2019 | Trade-off between Security and Capacity | IWT | Secure and robust against attacks | Time complexity for the proposed algorithm is high | BPP = 1.000<br>PSNR = 43.67<br>SSIM = 0.9546 |
| **RS87** | 2021 | Statistical Steganalysis Attacks | DCT | Robustness against statistical analysis attacks | Low relative embedding rate | BPP = 0.6000<br>SSIM = 0.9878<br>NC = 0.0987 |
| **RS88** | 2021 | Stego Image distortion | DCT | Robustness against RS attacks | Relatively low embedding capacity | BPP = 0.1000<br>PSNR = 43.45 |
| **RS93** | 2021 | Trade-off between Security and Capacity | IWT | Robust against universal steganalysis attacks with higher embedding capacity | High computational complexity | BPP = 5.25<br>PSNR = 44.58<br>SSIM = 0.9426 |
| **RS94** | 2021 | Low embedding capacity | IWT, CVD and LSB | Withstand steganalysis attacks and high embedding rate | Image distortion detected | BPP = 2.63<br>PSNR = 38.85 |
| **RS96** | 2021 | Trade-off between Security and Capacity | IWT | Achieves higher level of security | Time complexity is relatively higher | BPP = 1.000<br>PSNR = 51.83<br>SSIM = 0.9964 |
| **RS70** | 2022 | Trade-off between Security and Capacity | RDH, PVO and Prediction Error Histogram Shifting (PEHS) | Resist RS steganalysis and provide secure data transmission | Computational complexity is high for the implementation | BPP = 1.677<br>PSNR = 46.61<br>RS = 74% |
| **RS73** | 2022 | Low embedding rate and stego image distortion | RDHEI | Ensures losses data extraction | Distortion of image with higher embedding capacity | BPP = 0.4994<br>PSNR = 26.56<br>MSE = 0.3445 |

*(Continued)*

**Table 8.** (Continued)

| Reviewed Study (RP) | Year | Problem/Issue | Technique/ Method | Strength | Limitation | Evaluation Metric Results |
|---|---|---|---|---|---|---|
| **RS74** | 2022 | Statistical Steganalysis Attacks | RDH, Arnold Transform (AT), and DCT | High degree of robustness, imperceptibility, and visual quality of stego image | Low embedding rate | BPP = 1.000<br>RS = 0.0055<br>PSNR = 46.71<br>NC = 0.9944<br>SSIM = 0.9849 |
| **RS78** | 2022 | Stego image distortion | RNS | Boosts the anti-steganalysis capability | Low embedding rate | BPP = 0.4000 |
| **RS89** | 2022 | Statistical Steganalysis Attacks | DWT and Alpha Blending | High visual image quality and imperceptibility to withstand attacks | Low embedding capacity | BPP = 1.000<br>PSNR = 66.50<br>MSE = 0.1206 |
| **RS95** | 2022 | Statistical Steganalysis Attacks | IWT | Robustness against attacks with high imperceptibility | Embedding capacity not discussed | PSNR = 46.08<br>MSE = 0.5632<br>SSIM = 0.9900 |

avoid bias, the Delphi Expert Method [110] was adopted to evaluate the studies culminating in the rating provided for the robustness against attacks and overall security. All five researchers acted as experts and evaluated each study against the set of key parameters separately. Thereafter, a meeting was called to consolidate each rating. Where individual opinions differ, the cycle of Delphi was reinitiated until a consensus was reached. The method was designed in such a way that the researchers provided reasoning for individual responses. This was to help confirm the plausibility and strength of the individual researchers' evaluation.

## 4.6 Future scope and research directions for image steganography

The challenge of image steganography remains to achieve high embedding payload capacity while maintaining robustness, distortion resistance, imperceptibility, and overall security (undetectability). This challenge still exists in many of the reviewed works. The existing systems suffer from low embedding rate, low visual quality of stego image, image distortion, high computational complexity, performance accuracy, low throughput efficiency, as well as detection and modification of secret data. These gaps are largely due to the techniques employed by the existing works. Other identified gaps in most of the existing works are vulnerabilities such as double-frequencies, zero points, and non-accurate detection of statistical steganalysis results. These vulnerabilities have been extensively exploited by steganalysers.

Several of the reviewed works have no layer of protection against unauthorized access to secret data. This is because many of the existing works did not apply cryptographic protocols. Those that implemented cryptography for encryption and decryption are also based on the raster order LSB substitution method which is prone to RS statistical steganalysis attacks [111]. From Tables 10 and 11, only 34 out of the 125 reviewed papers employed encryption (cryptography). This represents 27% of all reviewed papers. The key aim of image steganography technique is to hide the existence of secret data using cover objects (audio, video, image, text, network) [112,113]. Also, for steganography to achieve its aim, the transferred message on the recipient side should be the same as the original message without noticeable suspicion by a third party [114,115]. Embedding secret data into the cover object does not provide the security needed [116–118]. This is because, an unauthorized person can read the message when the cover image is attacked, breaking the requirement for confidentiality of the message.

The analysis of the previous works has shown that there is a need to put in place appropriate corrective measures to strike an adequate balance between high payload and security against statistical steganalysis including RS attacks. Thus, techniques that achieve higher payload

**Table 9. Adaptive domain-based image steganography techniques.**

| Reviewed Study (RP) | Year | Problem/Issue | Technique/ Method | Strength | Limitation | Evaluation Metric Results |
|---|---|---|---|---|---|---|
| RS67 | 2013 | Stego Image Distortion | RDH | Higher visual image quality | Payload capacity is relatively low | BPP = 1.000<br>PSNR = 60.65<br>SSIM = 0.9813 |
| RS117 | 2013 | Low embedding capacity | Field Programmable Gate Array (FPGA) | High payload capacity and image quality | Time complexity of the application is high | BPP = 4.000<br>PSNR = 45.65<br>MSE = 0.4564 |
| RS8 | 2014 | Trade-off between Security and Capacity | GA | High Visual Image quality and high embedding capacity | Steganalysis attacks not simulated | BPP = 1.96<br>PSNR = 45.39 |
| RS9 | 2015 | Stego Image Distortion | GA, Logistics Maps and LSB | Attains high level of security with less computational time | Low embedding capacity | PSNR = 51.33<br>MSE = 0.0032<br>SSIM = 0.9997 |
| RS1 | 2016 | Statistical Steganalysis Attacks | GA | Increased payload capacity | Not robust against steganalysis attacks | PSNR mentioned but values not stated |
| RS3 | 2016 | Statistical Steganalysis Attacks | GA, LSB and AES | High image visual quality | Embedding capacity not discussed | PSNR mentioned but values not stated |
| RS7 | 2016 | Stego Image Distortion | GA and DCT | Less visual stego distortion | Robustness decreases with slight variation in pixel discontinuities | Capacity = 68.75%<br>PSNR = 52.78<br>MSE = 0.3428<br>NC = 0.9999 |
| RS27 | 2016 | Trade-off between Security and Capacity | CNN, AES and LSB | Stego image quality and High imperceptibility | Training model time is high | BPP = 3.00<br>PSNR = 40.41<br>SSIM = 0.7200 |
| RS28 | 2016 | Statistical Steganalysis Attacks | CNN, AES, LSB, and IWT | Improved image visual quality | Low embedding rate capacity | Capacity = 19%<br>PSNR = 59.51<br>MSE = 0.0728 |
| RS120 | 2016 | Stego Image Distortion | Content Adaptive, MiPOD and LSB-M | High un-detectability against universal statistical analysis | Image distortion noticed and low embedding capacity | BPP = 0.5000<br>RS = 1.234% |
| RS6 | 2018 | Statistical Steganalysis Attacks | GA and LSB | Increased imperceptibility and high capacity | Not Robust against certain attacks | PSNR = 63<br>RS = 6.25% |
| RS11 | 2018 | Statistical Steganalysis Attacks | GAN and CNN | High imperceptibility and security against attacks | Low embedding capacity | BPP = 0.5123 |
| RS29 | 2018 | Statistical Steganalysis Attacks | CNN | Possibility to detect corrupted cover image | Low embedding capacity and high training model time | Capacity = 19%<br>PSNR = 51<br>MSE = 0.4898<br>SSIM = 0.9998 |
| RS13 | 2019 | Stego Image Distortion | GAN | High robustness against statistical attacks | Low embedding rate | BPP = 0.4000 |
| RS30 | 2019 | Statistical Steganalysis Attacks | CNN | Better security performance against steganalyzer | Embedding payload capacity is relatively low | BPP = 0.5000 |
| RS31 | 2019 | Statistical Steganalysis Attacks | CNN and RDH | Robust against some statistical analysis | Low embedding payload capacity | BPP = 0.8<br>PSNR = 53.87 |
| RS38 | 2019 | Statistical Steganalysis Attacks | DL | High rate of invisibility | High model training and low embedding capacity | BPP = 0.500<br>PSNR = 32.17<br>MSE = 0.9832<br>SSIM = 0.9845 |
| RS2 | 2020 | Low visual quality | GA and RNS | Robust against steganalysis and cryptanalysis | Embedding capacity not discussed | PSNR = 13.0036<br>MSE = 0.3683 |
| RS14 | 2020 | Statistical Steganalysis Attacks | GAN | Improved security of adversarial images | Embedding rate and capacity not discussed | RS = 0.523<br>PSNR = 44.6 |
| RS32 | 2020 | Trade-off between Security and Capacity | CNN, LSB and Fuzzy Logic | Provided high embedding capacity | Distortion noticed with increased capacity | Capacity = 47.86%<br>PSNR = 45.87<br>MSE = 0.4536<br>SSIM = 0.8451 |

*(Continued)*

**Table 9.** (Continued)

| Reviewed Study (RP) | Year | Problem/Issue | Technique/ Method | Strength | Limitation | Evaluation Metric Results |
|---|---|---|---|---|---|---|
| RS35 | 2020 | Stego Image Distortion | CNN and LSB | Provide comprehensive resistance to steganalysis attacks | Embedding capacity was not discussed | PSNR = 50.73 MSE = 0.5494 |
| RS79 | 2020 | Statistical Steganalysis Attacks | RNS, Mobile edge computing and IoT | Maintains high visual image quality and resist steganalysis | Relatively low payload capacity | BPP = 0.05 PSNR = 82.75 MSE = 0.0003 SSIM = 1.000 |
| RS5 | 2021 | Stego Image Distortion | GA | Robust against steganalysis attacks | Low embedding capacity | BPP = 1 PSNR = 80.42 SSIM = 0.9988 |
| RS15 | 2021 | Statistical Steganalysis Attacks | GAN | High security level against single image steganalysis | Image distortion with appreciable level of capacity increase | BPP = 0.4000 RS = 1.200 |
| RS16 | 2021 | Statistical Steganalysis Attacks | GAN and Sparse Cover | High security improvement | Payload capacity limited | BPP = 0.5000 RS = 0.600 |
| RS17 | 2021 | Statistical Steganalysis Attacks | GAN | High visual image quality and improved security | Payload capacity not discussed | PSNR = 44.47 MSE = 2.550 SSIM = 0.9900 |
| RS19 | 2021 | Statistical Steganalysis Attacks | GAN | Improved security against CNN based steganalysis | Low embedding rate | BPP = 0.4000 |
| RS23 | 2021 | Statistical Steganalysis Attacks | GAN | High steganalysis security detection | High model training time | BPP = 0.400 PSNR = 35.67 |
| RS33 | 2021 | Stego Image Distortion | CNN and Vernam Algorithm | High image visual quality | Noticeable distortions with increased bit length | BPP = 2.923 PSNR = 55.07 MSE = 0.2023 SSIM = 0.9531 |
| RS39 | 2021 | Statistical Steganalysis Attacks | DL | High robustness against image modification | Run time efficiency of the algorithm is low | BPP = 0.800 |
| RS77 | 2021 | Statistical Steganalysis Attacks | RNS and CNN | High imperceptibility and improved security | Low embedding rate | BPP = 0.400 |
| RS4 | 2022 | Low visual quality | GA and IWT | High image visual quality and imperceptibility achieved | Payload capacity not good | BPP = 0.75 PSNR = 51.77 MSE = 0.4319 SSIM = 0.9968 |
| RS20 | 2022 | Statistical Steganalysis Attacks | GAN and CNN | High improvement in imperceptibility and detection rate | Embedding capacity is low | BPP = 0.4000 |
| RS21 | 2022 | Stego Image Distortion | GAN | High improvement in security and resistance against statistical attacks | Robustness decreases with increasing bit length | BPP = 0.5 PSNR = 27.60 MSE = 0.0023 SSIM = 0.9853 |
| RS22 | 2022 | Low visual quality | GAN | Robust against steganalysis attacks | The embedding capacity payload is low | BPP = 0.400 PSNR = 42.64 SSIM = 0.4984 |
| RS24 | 2022 | Statistical Steganalysis Attacks | GAN and Neural Style Transfer | Robust against stegoexpose than existing methods | High model training time | BPP = 1.000 PSNR = 43.95 SSIM = 0.9950 |
| RS25 | 2022 | Trade-off between Security and Capacity | GAN | Robustness and better security performance | The embedding capacity is very low | BPP = 0.400 |
| RS26 | 2022 | Statistical Steganalysis Attacks | GAN | Improves overall image system security and reduces loss of secret information | Distortion observed in stego image as payload increases further | BPP = 5.61 PSNR = 38.96 SSIM = 0.9800 |
| RS34 | 2022 | Low visual quality | CNN and Slice Encryption | More payload capacity and ability to withstand various attacks | Message length could easily be estimated | Capacity = 30225 bits PSNR = 55.48 MSE = 0.4322 SSIM = 0.9940 |

*(Continued)*

**Table 9.** (Continued)

| Reviewed Study (RP) | Year | Problem/Issue | Technique/ Method | Strength | Limitation | Evaluation Metric Results |
|---|---|---|---|---|---|---|
| **RS36** | 2022 | Trade-off between Security and Capacity | CNN and RDH | High embedding capacity with strong security features | High model training time | PSNR = 40.65<br>MSE = 0.0456<br>SSIM = 0.9800 |
| **RS37** | 2022 | Statistical Steganalysis Attacks | CNN and hash generation model | Better robustness and security | Inefficiency of searching the index database | BPP = 0.800 |
| **RS40** | 2022 | Statistical Steganalysis Attacks | DL | High security performance against modern steganalyzer | Learning stability is a bit lower comparatively | BPP = 0.500 |
| **RS10** | 2023 | Statistical Steganalysis Attacks | LSB, ECC and GA | Robust against RS statistical steganalysis attacks | High Embedding payload capacity | BPP = 3.39<br>MSE = 0.0999<br>PSNR = 50.53<br>SSIM = 0.9983<br>RS = 0.2450 |
| **RS12** | 2023 | Low embedding capacity | Hamilton Path, GA | High robustness against attacks | High embedding capacity | BPP = 3<br>PSNR = 41.80 |
| **RS18** | 2023 | Statistical Steganalysis Attacks | GA, LSB | Robustness against attacks | High-Capacity payload | BPP = 3.5<br>PSNR = 46.07<br>SSIM = 0.9979 |

capacity and better-corrected pixels in ensuring enhanced security protection of secret data in storage and transmission are required. One key challenge of the image steganography embedding process is the secret message size [119]. This challenge could be overcome by employing

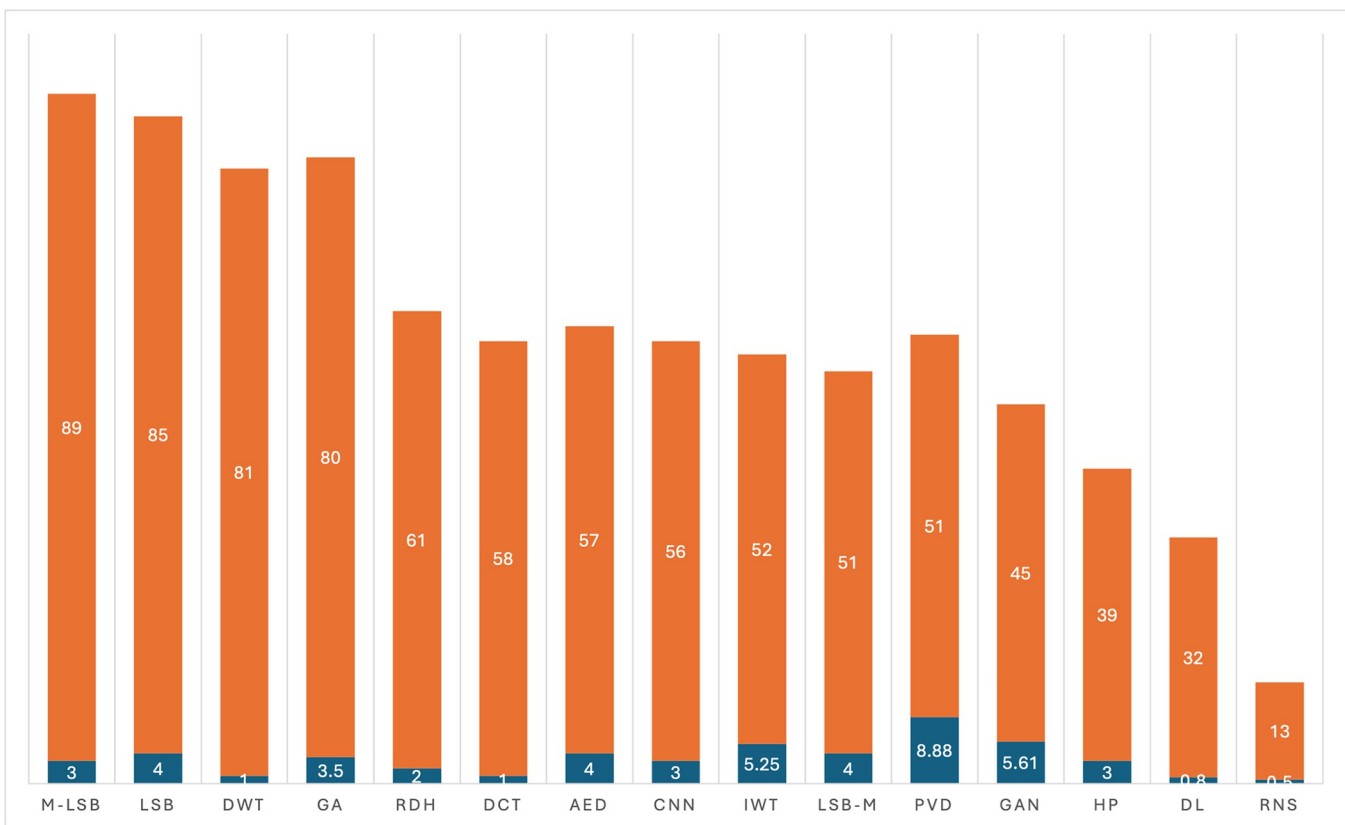

**Fig 7. Comparison of embedding capacity and security of image steganography techniques.**

**Table 10. Comparison of various existing image steganography techniques and methods for grayscale images.**

| Reviewed Paper (RS) | Dataset Used | Embedding Process | Data Embedding | Secret Image Type | Real-time | Encryption? | Compression? | PSNR (dB) | Robustness Against Attacks | Security |
|---|---|---|---|---|---|---|---|---|---|---|
| RS3 | Lena | Adaptive | Random | Gray | No | Yes | Yes | N/A | Medium | Average |
| RS5 | Lena, Baboon, Peper, Lake | Adaptive | Random | Gray | Yes | No | No | 80.42 | Medium | Good |
| RS9 | Lena, Lion | Adaptive | Random | Gray | Yes | No | No | 51.33 | Medium | Average |
| RS12 | Lena | Adaptive | Random | Gray | Yes | Yes | No | 41.8 | Medium | Average |
| RS13 | Humanface | Adaptive | Random | Gray | Yes | No | Yes | N/A | Low | Average |
| RS14 | Building | Adaptive | Random | Gray | Yes | No | No | 44.6 | Medium | Low |
| RS15 | Building | Adaptive | Random | Gray | Yes | No | No | N/A | Low | Average |
| RS16 | Road, Building | Adaptive | Random | Gray | Yes | No | No | N/A | Low | Average |
| RS18 | Lena, Pepper, Baboon, Cameraman | Adaptive | Random | Gray | Yes | No | No | 46.07 | Medium | Average |
| RS19 | Building | Adaptive | Random | Gray | Yes | No | No | N/A | Low | Low |
| RS20 | Building | Adaptive | Random | Gray | Yes | No | No | N/A | Medium | Low |
| RS23 | Building | Adaptive | Random | Gray | Yes | No | No | 35.67 | Medium | Low |
| RS26 | Building | Adaptive | Random | Gray | No | No | No | 38.96 | Medium | Average |
| RS30 | Building | Adaptive | Random | Gray | Yes | No | No | N/A | Medium | Low |
| RS31 | Dog, Puppy, Laptop | Adaptive | Random | Gray | Yes | No | No | 53.87 | Medium | Average |
| RS32 | Lena, Lion Snow, Aeroplane | Adaptive | Random | Gray | Yes | Yes | No | 45.87 | High | Average |
| RS33 | Lion | Adaptive | Random | Gray | No | Yes | No | 55.07 | High | Average |
| RS34 | Lena, Coins, Baboon, Cameraman | Adaptive | Random | Gray | Yes | Yes | No | 55.48 | High | Average |
| RS35 | Lena, Lion, Cameraman | Adaptive | Random | Gray | Yes | No | No | 50.73 | Medium | Average |
| RS40 | Building | Adaptive | Random | Gray | Yes | No | No | N/A | Medium | Low |
| RS41 | Lena, Cameraman, Pirates | Spatial | Sequential | Gray | No | No | No | 39.39 | Low | Average |
| RS42 | Imgaeset | Spatial | Sequential | Gray | Yes | No | No | 52.52 | Medium | Average |
| RS43 | Lena, Tiffany, Baboon, Jet, Bird, Castle, Pepper, Boat | Spatial | Random | Gray | Yes | Yes | No | 37.66 | High | Average |
| RS44 | Lena, Tiffany House, Milk, Jet | Spatial | Sequential | Gray | No | No | Yes | 41.58 | Medium | Average |
| RS45 | Lena, House | Spatial | Sequential | Gray | No | No | No | N/A | Low | Low |
| RS46 | Lena, Pepper, Jet, Airplane Truck, Tank, Baboon, Boat | Spatial | Sequential | Gray | Yes | Yes | Yes | 36.38 | High | Average |
| RS49 | Imageset | Spatial | Random | Gray | Yes | No | No | N/A | Low | Low |
| RS50 | Lena, Couple, Baboon, Boat Pepper, Man, Tiffany, | Spatial | Sequential | Gray | Yes | No | No | 33.83 | Low | Low |
| RS51 | Lena, Couple, Baboon, Boat Pepper, Man, Tiffany, baby | Spatial | Sequential | Gray | Yes | No | No | 39.09 | Low | Average |
| RS52 | Imgaeset | Spatial | Sequential | Gray | Yes | No | No | N/A | Low | Low |
| RS53 | Lena, Baboon | Spatial | Random | Gray | No | No | No | 40.79 | Low | Average |
| RS54 | Building | Spatial | Random | Gray | Yes | No | No | N/A | Low | Low |
| RS55 | Lena, Couple, Baboon, Boat Pepper, Man Tiffany, | Spatial | Random | Gray | Yes | Yes | No | 40.95 | High | Average |
| RS56 | MRI Image | Spatial | Random | Gray | Yes | Yes | Yes | 50.39 | High | Good |

*(Continued)*

**Table 10.** (Continued)

| Reviewed Paper (RS) | Dataset Used | Embedding Process | Data Embedding | Secret Image Type | Real-time | Encryption? | Compression? | PSNR (dB) | Robustness Against Attacks | Security |
|---|---|---|---|---|---|---|---|---|---|---|
| RS57 | Airplane, Baboon | Spatial | Random | Gray | Yes | No | Yes | 43.62 | High | Average |
| RS58 | Building | Spatial | Random | Color | Yes | No | No | 57.33 | Medium | Average |
| RS59 | Lena, Couple, Baboon, Boat Pepper, Man, Tiffany, baby | Spatial | Sequential | Gray | Yes | No | No | 35.68 | Low | Low |
| RS60 | Buildings | Spatial | Random | Gray | Yes | No | No | 48.59 | Low | Average |
| RS61 | Baboon Cameraman Airplane Goldhill Lena Peppers Tiffany Boat Aerial Clown Zelda | Spatial | Random | Gray | Yes | Yes | No | 51.68 | Medium | Average |
| RS63 | Baboon, Pepper, Airplane | Spatial | Random | Gray | Yes | No | No | 48.61 | Medium | Average |
| RS64 | Bossbase | Spatial | Random | Gray | Yes | No | No | 57 | Medium | Average |
| RS66 | Lena | Spatial | Sequential | Gray | Yes | No | No | 50.51 | Medium | Average |
| RS67 | Lena, Couple, Baboon, Boat Pepper, Man, Tiffany, baby | Adaptive | Sequential | Gray | No | No | No | 60.65 | Medium | Medium |
| RS68 | Imageset | Transform | Random | Gray | Yes | No | No | 47.64 | Low | Average |
| RS69 | Lena, Couple, Baboon, Boat Pepper, Cameraman, Tiffany | Spatial | Random | Gray | Yes | Yes | No | 31.99 | High | Average |
| RS70 | Lena, Boat, Pepper, Barbara, Goldhill | Transform | Sequential | Gray | No | No | No | 46.61 | Low | Average |
| RS71 | Lena, Couple, Baboon, Boat Pepper, Cameraman, Tiffany | Spatial | Sequential | Gray | No | No | No | 43.13 | Low | Average |
| RS72 | Lena, F16, Boat, Zelda Pepper, Lake, Barbara, Baboon | Spatial | Random | Gray | Yes | Yes | Yes | N/A | High | Average |
| RS73 | Lena, Zelda Couple, Boat, Pepper, Elaine, Lake Baboon | Transform | Random | Gray | Yes | Yes | No | 25.56 | High | Average |
| RS76 | Glasscup, Statue | Transform | Sequential | Gray | Yes | No | No | N/A | Low | Low |
| RS77 | Building | Adaptive | Random | Gray | Yes | No | No | N/A | Low | Low |
| RS78 | Building | Transform | Random | Gray | Yes | No | No | N/A | Low | Low |
| RS79 | Deer, Boat, Cameraman | Adaptive | Random | Gray | Yes | No | No | 82.75 | Medium | Good |
| RS82 | Building | Transform | Random | Gray | Yes | Yes | No | 36.68 | High | Average |
| RS83 | Lena, House, Baboon, Lily Flowers | Transform | Sequential | Gray | No | Yes | No | N/A | Low | Low |
| RS84 | Lena, Boat, Baboon, House, Woman, Pepper | Transform | Sequential | Gray | Yes | No | No | 43.97 | Low | Average |
| RS88 | Lena | Transform | Random | Gray | Yes | No | Yes | 43.45 | High | Average |
| RS90 | Imageste | Transform | Sequential | Gray | Yes | Yes | Yes | 35.06 | High | Average |
| RS91 | Lena, Pepper, Baboon, Boat | Transform | Sequential | Gray | Yes | No | Yes | 32.44 | Medium | Average |
| RS92 | Lena, Tank, Elaine, Boat Baboon, Couple, Airplane | Transform | Sequential | Gray | Yes | No | No | 43.67 | Low | Average |
| RS93 | Baboon | Transform | Sequential | Gray | No | No | No | 44.58 | Low | Average |
| RS94 | Lena, Woman | Transform | Random | Gray | Yes | No | No | 38.85 | Low | Average |
| RS95 | Lena | Transform | Random | Gray | Yes | No | No | 46.08 | Low | Average |
| RS96 | Lena, Woman, Baboon, Gun Aeroplane, Man, Portrait | Transform | Random | Gray | Yes | No | No | 51.83 | medium | Average |
| RS97 | Chinese, Lena, English Baboon | Spatial | Random | Gray | Yes | No | No | 11.96 | Low | Low |
| RS98 | Lena, Pepper, Cameraman Baboon | Spatial | Sequential | Gray | No | No | No | 49.40 | Low | Average |

(*Continued*)

**Table 10.** (Continued)

| Reviewed Paper (RS) | Dataset Used | Embed-ding Process | Data Embed-ding | Secret Image Type | Real-time | Encry-ption? | Comp-ression? | PSNR (dB) | Robustness Against Attacks | Security |
|---|---|---|---|---|---|---|---|---|---|---|
| RS99 | Lena, Baboon, Mandrill, Boat Barbara, Zelda | Spatial | Sequential | Gray | No | No | No | 51.14 | Medium | Average |
| RS100 | Lena, Baboon Boat, Clown Zelda | Spatial | Sequential | Gray | Yes | No | No | 51.16 | Medium | Average |
| RS101 | Lena, Pepper, Boat, Goldhill F16, Baboon | Spatial | Sequential | Gray | Yes | No | No | 31.69 | Low | Low |
| RS105 | Baboon, Aeroplane | Spatial | Random | Gray | Yes | Yes | No | 44.09 | High | Average |
| RS109 | Imageset | Spatial | Sequential | Gray | No | No | No | N/A | Low | Low |
| RS110 | Baseboss | Spatial | Sequential | Gray | Yes | No | No | N/A | Medium | Low |
| RS112 | Flower, Lena, Rabbit, Garden | Spatial | Sequential | Gray | Yes | Yes | No | 64.15 | High | Good |
| RS113 | Building | Spatial | Random | Gray | Yes | Yes | No | N/A | Low | Low |
| RS116 | Baboon, Barbara, House, | Spatial | Sequential | Gray | Yes | Yes | Yes | 40.09 | High | Average |
| RS117 | Boat | Adaptive | Random | Gray | Yes | No | No | 45.65 | Low | Average |
| RS118 | Lena, House Couple, Boat, Truck, Pepper, Female, Lake, Male Baboon, Splash, Cameraman | Spatial | Random | Gray | Yes | No | No | 36.45 | Medium | Low |
| RS120 | Hill, Chapel | Adaptive | Sequential | Gray | Yes | Yes | No | N/A | High | Average |
| RS121 | Lena | Spatial | Random | Gray | Yes | No | No | 45.05 | Low | Average |
| RS123 | Church, man | Spatial | Random | Gray | Yes | No | No | N/A | Low | Low |
| RS125 | Imageset | Spatial | Random | Gray | Yes | No | No | 40.12 | Low | Average |

lossless compression algorithm techniques to achieve higher payload capacity and high embedding rate [120]. From Table 10, only 13% of all the reviewed articles in this study implemented data compression. Compression reduces the secret data size before embedding process begins [121,122].

Clearly, this systematic literature review has shown that the research direction in image steganography has been broad and diverse since 2012. As challenges in image steganography continue, the research domain also continues to evolve. Aside from the traditional methods, researchers have begun experimenting other areas of application for image steganography. For example, Table 4 shows that 9 of the papers adopted other different techniques than the known traditional methods for steganography. This can be inferred that scholars within the image steganography domain are exploring newer and more innovative approaches.

Future research directions could enhance the security and robustness of image steganography applications by:

- Cryptographic protocols as a layer of security protection. Higher security and robustness in image steganography can be achieved using multiple encryptions to mask and scramble the content of the secret message before embedding. Encrypted embedded secret data have more ability to resist steganalysis.

- Future research could explore compression and image enhancement techniques to achieve a high payload while maintaining image visual quality. This could help solve the problem of balancing the tradeoff between security and embedding capacity

- Future research could utilize other novel techniques from domains that have the propensity to achieve computationally efficient, reduced computational complexity, improved

**Table 11. Comparison of various existing image steganography techniques and methods for color images.**

| Reviewed Paper (RS) | Dataset Used | Embedding Process | Data Embedding | Secret Image Type | Real-time | Encry-ption? | Comp-ression? | PSNR (dB) | Robustness Against Attacks | Security |
|---|---|---|---|---|---|---|---|---|---|---|
| RS1 | Baby, Pigeon, Flower | Adaptive | Random | Color | No | No | No | N/A | Low | Low |
| RS2 | Lena, pepper | Adaptive | Random | Color | Yes | Yes | No | 13.0036 | Medium | Low |
| RS4 | Lena, Pepper, Baboon | Adaptive | Sequential | Color | No | No | No | 51.77 | Medium | Average |
| RS6 | Paper | Adaptive | Random | Color | No | No | No | 63 | Medium | Avera |
| RS7 | Monkey, Flower | Adaptive | Random | Color | No | No | No | 52.78 | Low | Average |
| RS8 | Lena, Pepper, Aeroplane, Baboon | Adaptive | Random | Color | No | No | No | 45.39 | Low | Average |
| RS10 | Lena | Adaptive | Random | Color | Yes | Yes | Yes | 50.53 | High | High |
| RS11 | Humanface | Adaptive | Random | Color | Yes | No | No | N/A | Medium | Average |
| RS17 | Bridge | Adaptive | Random | Color | No | No | No | 44.47 | Low | Average |
| RS21 | Flowers, Frog | Adaptive | Random | Color | Yes | No | No | 27.60 | Medium | Low |
| RS22 | Bird, Humanface | Adaptive | Random | Color | Yes | No | No | 42.64 | Medium | Average |
| RS24 | Imagenet | Adaptive | Random | Color | Yes | No | No | 43.95 | Medium | Average |
| RS25 | Woman | Adaptive | Random | Color | Yes | Yes | No | N/A | Low | Average |
| RS27 | Flower, baby | Adaptive | Radom | Color | Yes | Yes | No | 40.41 | Medium | Average |
| RS28 | Woman | Adaptive | Random | Color | Yes | Yes | No | 59.51 | High | Average |
| RS29 | Lena | Adaptive | Random | Color | Yes | No | No | 51 | Medium | Average |
| RS36 | Imageset | Adaptive | Random | Color | No | No | No | 40.65 | Low | Average |
| RS37 | Seabird | Adaptive | Random | Color | Yes | Yes | No | N/A | Medium | Low |
| RS38 | Wordnet | Adaptive | Random | Color | Yes | Yes | No | 32.17 | Medium | Low |
| RS39 | Baby with Piano | Adaptive | Sequential | Color | No | No | No | N/A | Low | Low |
| RS47 | Lena, Strawberry | Spatial | Sequential | Color | Yes | No | No | 47.99 | Low | Average |
| RS48 | Lena | Spatial | Sequential | Color | Yes | No | No | 25 | Low | Low |
| RS58 | Building | Spatial | Random | Color | Yes | No | No | 57.33 | Medium | Average |
| RS62 | Imageset | Spatial | Random | Color | Yes | No | No | N/A | Low | Low |
| RS65 | Imageset | Spatial | Random | Color | No | No | No | 46.89 | Low | Average |
| RS74 | Pepper, Boat Baboon, Aeroplane | Transform | Random | Color | Yes | No | No | 46.71 | Low | Average |
| RS75 | Lena, Pepper | Spatial | Sequential | Color | Yes | No | No | 51.93 | Medium | Average |
| RS80 | Lena | Transform | Random | Color | Yes | No | No | 58.95 | Medium | Average |
| RS81 | Sea, Grass | Transform | Sequential | Color | Yes | No | No | 81.33 | Medium | Good |
| RS85 | Lena, Baboon Pepper | Transform | Random | Color | Yes | No | Yes | 53.38 | High | Average |
| RS86 | House, Toy Man | Transform | Sequential | Color | Yes | No | No | N/A | Low | Low |
| RS87 | Bird | Transform | Random | Color | Yes | No | No | N/A | Low | Low |

(*Continued*)

**Table 11.** (Continued)

| Reviewed Paper (RS) | Dataset Used | Embed-ding Process | Data Embed-ding | Secret Image Type | Real-time | Encry-ption? | Comp-ression? | PSNR (dB) | Robustness Against Attacks | Security |
|---|---|---|---|---|---|---|---|---|---|---|
| RS89 | Lena, Pepper Cameraman, Baboon | Transform | Random | Color | Yes | Yes | Yes | 66.50 | High | Good |
| RS102 | Lena, Apple Airplane, Baboon, | Spatial | Sequential | Color | Yes | No | No | 56.44 | Medium | Average |
| RS103 | Imageset | Spatial | Random | Color | No | Yes | No | 74.02 | High | Good |
| RS104 | Butterfly | Spatial | Sequential | Color | No | No | Yes | 54.08 | Medium | Average |
| RS105 | Baboon, Aeroplane | Spatial | Random | Gray | Yes | Yes | No | 44.09 | High | Average |
| RS106 | Sea, Cow, Tree, House Church, | Spatial | Sequential | Color | Yes | No | Yes | 48.24 | Medium | Average |
| RS107 | Lena, Pepper, Baboon | Spatial | Random | Color | Yes | Yes | Yes | 89.03 | High | Good |
| RS108 | Lena, Baboon Aeroplane, Girl | Spatial | Random | Color | No | No | No | 83.27 | Medium | Good |
| RS111 | Baboon, building, Woman | Spatial | Random | Color | Yes | Yes | No | 85.664 | Medium | Average |
| RS114 | Lena, Apple, Butterfly Church, Orange | Spatial | Random | Color | No | Yes | No | 48.35 | High | Average |
| RS115 | Deer | Spatial | Random | Color | No | Yes | No | 49.56 | High | Average |
| RS119 | House, Lake Pepper, Baby, Baboon, Image1, | Spatial | Random | Color | Yes | Yes | Yes | 83.99 | Medium | Good |
| RS122 | Lena, Baboon, Pepper, man | Spatial | Random | Color | Yes | No | No | 69.45 | Medium | Average |
| RS124 | Man | Spatial | Random | Color | Yes | Yes | No | 30.30 | High | Average |

performance, and undetectability which are the major issues advocated for by researchers within the image steganography domain. For instance, imperceptibility and security could be improved by employing emerging technologies such as Blockchain Technology [123]. Stego-images containing secret data are often transmitted over unsecured public networks, thereby making the secret data susceptible to many attacks including man-in-the-middle attacks, tampering, and eavesdropping [124,125].

- Blockchain technology could be employed in image steganography to ensure stego-images are more secure and authenticated [114]. This is because, blockchain has immutable properties, easy traceability, tracking capabilities, and transparency [50]. In addition, future research could rely on emerging artificial intelligence and machine learning power technologies such as ChatGPT to provide robust techniques against steganographic attacks.

## 5. SLR results discussion and implication

The review focused on providing evidence on image steganography techniques that have been designed to resist statistical steganalysis attacks. The review has shown that several such techniques and methods, with the capability to withstand complex attacks, exist. This systematic literature review was based on key questions that provided a foundation for the review. The SLR results are provided as summarized answers to the study's research questions. **Table 12** provides the questions and a summary of the systematic literature review results.

### 5.1 Research trends in image steganography techniques

The review reveals an interesting result for image steganography research. Intriguingly, research on image steganography is skewed in terms of publication trends. The skewness in

**Table 12. Answers to SLR questions and summary of review results.**

| Item | Research Questions (RQ) | Answers to Research Questions |
|---|---|---|
| RQ1 | Q1. What have been the Trends in Publication of Image Steganography Applications? | The review of all the articles revealed an interesting result for image steganography research. Intriguingly, research on image steganography is skewed in terms of publication trend. The skewness in the publication trend for image steganography can be seen in analysis concerning the year of publication, publication outlets, country of origin of corresponding author, and application domains for image steganography. More than 50% of articles were published after 2020. IEEE Explore is the most preferred destination for scholars researching image steganography, while majority of the articles emanated from India and China with no single article from Sub-Saharan Africa (SSA), indicating that research in the field of steganography is low in Africa. |
| RQ2 | Q2. What Methods and Techniques are Used in Image Steganography for Resisting Statistical Attacks? | After reviewing the articles, Generative Adversarial Networks (GAN) was observed as the most preferred image steganography technique, and machine learning based algorithms such as DL, CNN, and GA have dominated image steganography research. The results revealed that adaptive methods are overtaking spatial and transform domain approaches. Previously preferred traditional techniques such as LSB, PVD, DCT and IWT algorithms are receiving less attention in image steganography research and applications. |
| RQ3 | Q3. What are the Standard Performance Evaluation Metrics for Image Steganography Techniques | The review of all the articles revealed several performance metrics that have been used to evaluate image steganography techniques. Most of the articles used more than one performance of evaluation metrics. Five performance evaluation metrics were observed to be commonly used by majority of the studies reviewed. These metrics are PSNR, MSE, SSIM, NC, and BPP. Few of the articles did not discuss performance evaluation metrics. |
| RQ4 | Q4 What Security Impact Has the Techniques have on Image steganography for Resisting Statistical Attacks? | The reviewed articles show that four benchmark datasets consisting of BOSS base datasets, USC-SIPI, Seam Carving Original Q75, and 24 KODAK image Databases have widely been used. Adaptive embedding techniques such as GAN, GA and CNN were resistant to geometrics attacks, and statistical detection analysis attacks such as RS and Histogram analysis attacks. The visual quality of adaptive based methods and undetectability of secret message were high and robust against noise cropping and less prone to image rotation. However adaptive methods have limited embedding capacity. However, even though spatial domain techniques such as LSB, LSB-M, PVD have high embedding capacity and visual quality, they are highly prone to noise cropping, rotation, non-structural detection analysis and statistical detection analysis attacks such as RS and Histogram analysis attacks. Spatial domain techniques are also vulnerable to geometric attacks. Transform domain techniques such as DCT and IWT offered high security consideration than spatial domain methods but less effective when compared to adaptive embedding methods. Only few of the techniques have also been implemented in a real-time application. |

(*Continued*)

**Table 12.** (Continued)

| Item | Research Questions (RQ) | Answers to Research Questions |
|---|---|---|
| RQ5 | Q5. What are the Future Scope and Research Direction for Image Steganography? | The review has shown that research direction in image steganography have been broad and divergent since 2012. As challenges in image steganography continue, research domain also continues to evolve. Aside the traditional methods, researchers have begun experimenting other areas of application for image steganography. The challenge of image steganography remains achieving high embedding payload capacity while maintaining robustness, distortion resistance, imperceptibility, and overall security (un-detectability). It is therefore recommended that researcher may consider emerging technologies such as blockchain technology, artificial neural networks, encryption, and compression in future research works to improve security and embedding capacity. |

the publication trends for image steganography can be seen in analysis concerning the year of publication, publication outlets, country of origin of the corresponding author, and application domains for image steganography. The research shows that despite the growing interest in the research field in image steganography, research in the area took a sharp nosedive in 2020, but rather experienced astronomical expansion from 2021 to 2023. Approximately half of the papers studied in this research were published from 2021 to 2023. Indeed, cyber-attacks on organizations and individual data due to inherent vulnerabilities in network security protection were expanding [126], even before 2020. This might have contributed to the interest of researchers in this domain to find solutions to the ever-increasing threat. The volume of research conducted in this domain post-COVID-19 is not surprising, as the Coronavirus (COVID-19) pandemic resulted in an increased number and range of cyber-attacks resulting in personal and organizational data breaches and compromises [127]. The exponential increase in the research domain could be a direct response to the increasing trend of cyber-attacks during the COVID and the need for companies to work remotely as a means of cutting costs and making use of investments in technology during the pandemic. In terms of publication outlets, it is interesting to note that more than half of the articles reviewed were published in IEEE. The implication is that IEEE has become the destination of choice for researchers publishing studies on image steganography. This finding corroborates the study of Kaur et al., [50] where most of the reviewed papers were also published in IEEE. This brings to the fore the need to address the dominance in the publication of such crucial research areas by a particular publication house and expand the domain in other publication outlets. Although there are several publication outlets that publish research on image steganography, such outlets were dully not represented in this study. Given that image steganography techniques for resisting steganalysis have become a growing area of research interest, other publication outlets may put in place measures to attract researchers. This could include special issues concerning the domain and putting in place incentives to attract researchers. Surprisingly, despite the growing cases of cybercrimes in Sub-Sahara Africa [128], the interest of researchers in this geographic location is low. It must, however, be mentioned that researchers in Sub-Saharan Africa have begun showing interest in publishing in this area, as evidenced by a recent publication [70]. Developing research capabilities including collaboration with external scholars particularly those in India and China could ameliorate the low level of research by African scholars in this domain. The digital divide in Africa is growing. Internet penetration in Africa is also

expanding, and as a result, digital crimes have increased. Developing research capabilities and acquiring the requisite technical knowledge to research image steganography techniques could prevent many of the data breaches and cyber-attacks as well as save African-based organizations from data breaches and compromises.

## 5.2 Image steganography techniques for resisting steganalysis

As observed in Fig 6 and Table 4, Generative Adversarial Neural Networks (GAN) is the most preferred image steganography technique for resisting steganalysis attacks. This finding supports arguments by Liu et al., [129] that GAN has seen increasing achievement in the field of image steganography, computer vision, and natural language processing. From the review, the application of GAN in image steganography witnessed exponential growth between 2018 and 2022. GAN was first proposed in 2014 [130] and has seen great application in many fields of Computer Science. In image steganography, it improves security by resisting cover modification, enhances the cover selection and synthesis processes, and achieves overall security protection against steganalysis attacks. The security capabilities of GAN are higher than other adaptive methods and traditional spatial and transform domain methods [131]. Quite interestingly, despite the complexity associated with GAN-based image steganography approaches, the technique has seen overwhelming applications. The increase in the use of GAN processes is attributed to recent developments in deep learning-based steganalysis [132–134]. GAN has the capability to resist state-of-the-art deep learning-based steganalysis [135]. GAN also can be used to improve the security performance of image steganography techniques in other domains including spatial domain applications. These capabilities make GAN a considerable option for image steganography regardless of the complexity associated with it.

The study further shows that machine learning-based algorithms are recently dominating image steganography research. This confirms the argument by Hussain et al., [82] on the growth of machine learning techniques including GAN, DL, CNN, and GA. These machine learning-based algorithms have emerged as powerful tools for image steganography capable of resisting steganalysis attacks. Subramanian et al, [131] argue that machine learning-based algorithms will continue to see greater applications in future image steganography works. DL, GA, and CNN like other machine learning algorithms including GAN are great techniques for fooling steganalysis and preventing them from detecting secret images hiding in cover images. In addition to machine learning-based algorithms, the study reveals that researchers are exploring many other areas of application for image steganography. At least 9 of the reviewed articles were based on other methods rather than known traditional steganography methods or machine learning methods.

The overall implication is that previously preferred image steganography techniques particularly the least significant bit (LSB) insertion algorithms are becoming unpopular among data protection and information security researchers. This finding supports the assertion by Subramanian et. al., [131] that traditional algorithms like LSB are now receiving less attention in image steganographic applications. Between the spatial domain and transform domain, algorithms based on the spatial domain were more. This finding supports arguments by Hussain et al., [82] that the spatial domain methods for secret data embedding are more popular than the transform domain due to the easiness of embedding and extraction of data in the spatial domain. The spatial domain however suffers from less robustness. The major spatial domain methods include LSB, LSB-M, AED, PVD, and PH. The major transform domain methods identified were DCT and IWT techniques such as RDH and RNS however saw application across the various embedding domain processes (ie spatial, transform, and adaptive domains). Indeed, LSB is considered the fundamental and conventional steganography method capable

of hiding a larger secret message in a cover image without noticeable visual distortions. Over time, different variations of LSB have been developed. The disadvantage of LSB is that an increase in payload reduces the overall visual quality making it an easy target for attacks. Given the challenges of LSB, Wu and Tsai [136] proposed PVD using the difference between two neighboring pixels to determine the number of secret bits to be embedded. Since then, many steganographic methods have been proposed to improve the initial PVD method. From the study, it can further be observed that AED is one of the prominent embedding strategies in the spatial domain. AED schemes have the capability to maintain minimum visual quality and are noted to provide higher imperceptibility when compared to other spatial domains [137]. From Table 4, AED recorded the second highest techniques for image steganography. Different hybrid edge-based methods including combining canny edge and fuzzy edge adaptors [138,139] were observed in the articles reviewed for this study. The study has revealed varied techniques for protecting data against attacks. However, more research investigations are required to identify how emerging technologies including artificial neural networks (ANN) could be explored to provide harmonized security capabilities against statistical steganalysis attacks.

## 5.3 Security performance of image steganography against attacks

The systematic review results revealed that the most significant contribution of steganography techniques is resistance against statistical detection analysis attacks such as Regular-Singular (RS) and Histogram analysis attacks. Adaptive embedding techniques such as GAN, GA, and CNN and transform domain techniques including DCT and IWT methods were hard to expose to such statistical detection analysis attacks. However, spatial domain techniques including LSB and PVD were easy to expose. Most of the studies reviewed reported improvement against RS and histogram analysis attacks, indicating continued research improvement in overcoming these types of attacks. Another key significance of existing steganographic techniques is resistance against non-structural detection attacks. Machine learning-based algorithms proved difficult to detect by non-structural detection attacks, whereas spatial domain and transform domain methods were easily detectable. In terms of geometric attacks, it was observed that adaptive embedding techniques such as GAN and CNN and techniques-based transform domain methods were resistant and hard to geometric attacks while spatial domain methods were vulnerable to such attacks.

The visual quality of adaptive-based methods and the undetectability of secret messages were high and robust against noise cropping and less prone to image rotation. However adaptive methods have limited embedding capacity. Even though spatial domains such as LSB, LSB-M, and PVD have higher payload capacity and visual quality, they are highly prone to noise cropping, and rotation. Overall, most of the reviewed studies reported higher SI visual quality, an important measure in ensuring the transmission of secret data is not detectable by the HVS. Transform domains such as DCT and IWT offered higher security considerations than spatial domain methods but were less effective when compared to adaptive embedding methods. Only a few of the techniques have also been implemented in a real-time application. When evaluation of image steganography is done using capacity, traditional embedding algorithms including the various variations of LSB offer higher embedding capacity than machine learning-based techniques such as CNN, GAN, and DL.

Despite the notable progress achieved in image steganographic techniques, computational complexity and time complexity were observed to be a major challenge in all the reviewed papers. Even though computational complexity is a generic challenge as most studies indicated, adaptive embedding techniques such as CNN, DL, and GAN were reported to have

higher computational complexity results than both spatial domain and transform domain methods. This finding is, however, not surprising given that most of the adaptive embedding approaches were based on machine learning techniques. This is because, one key challenge associated with machine learning algorithms has been identified to be computational complexity [140,141]. The challenge of computational complexity is noted to significantly have a direct impact on image steganography techniques with respect to computational speed thereby having a tremendous impact on the performance of emerging image steganography applications. This notwithstanding, recent studies have reported measures to improve the computational complexity and time accuracy of machine learning algorithms [142]. This has occasioned the growing use of genetic algorithms (GA) in image steganography applications [70], as GA has been noted as reducing the computational complexities of machine learning-based algorithms.

From Tables 10 and 11, the results from the systematic review analysis have shown the positive effects of combining steganography and cryptography. The analysis further shows that image steganography studies that had implemented cryptography were rated high for robustness and good for overall security. The combined effects of cryptography and steganography provide an additional layer of protection for the privacy system against many security attacks [143,144]. Although the combination is noted as an extra payload on the time and space complexities of the application, it offers comparative advantages in terms of robustness, confidentiality, and privacy [145]. However, several techniques have recently been introduced to reduce the computational cost performance associated with the art of combining steganography and cryptography.

From Fig 7, Modified Least Significant Bits (M-LSB) had the highest PSNR value indicating the highest imperceptibility. This was obtained for RS 108. This was followed by RS111 with a PSNR value of 85, which utilized the LSB technique. For embedding capacity, the highest capacity recorded among the reviewed articles was 8.88BPP for the PVD technique. This was obtained in RS48. This was followed by RS26, a generative adversarial network (GAN) which obtained 5.61BPP. A careful examination of Tables 7–9 shows that Spatial domain techniques recorded the highest imperceptibility outcome. However, spatial domains are susceptible to steganalysis attacks. The average highest embedding capacity was recorded in the spatial domain and transform domain techniques. Genetic Algorithm (GA) and GAN applications under the adaptive domains showed the best results for balancing embedding capacity and robustness. This explains the growing use of GAN and GA algorithms. Even though, the high-capacity trade-off to security and robustness improvement remains a challenge [146–148], GAN, GA, and other emerging technologies such as generative artificial intelligence (AI) have the potential to overcome the challenge.

## 6. Conclusion, research validity, and limitation

The paper provided a systematic literature review of image steganography techniques that can withstand statistical steganalysis attacks. To the best of the Authors' knowledge and understanding of the existing literature, this systematic review is the first to have considered the entire spectrum of image steganography methods and techniques and their application in resisting steganalysis attacks. The study sampled 125 articles from four reputable electronic databases comprising ACM, IEEE, Science Direct, and Wiley. Using PRISMA for literature mapping, the articles were synthesized and analyzed using quantitative and qualitative methods. Trends in publication, techniques and methods, performance evaluation metrics, and the security impact of image steganography techniques against steganalysis were discussed. Extensive comparisons were drawn among existing techniques to evaluate their merits and limitations. Various future research directions in image steganography have been provided to help

researchers who may want to consider emerging technologies to enhance data protection and security.

Research validity is an important component in all studies, as biases have the potential to negatively impact the study outcome. The possible biases and the threat to the validity of this research emanate from the potential omission of articles in the selection and data extraction processes. Various databases and journals publish research on cryptography and steganography, which may contain relevant articles that meet the inclusion criteria for the study. However, the article selection was limited to four databases only. It therefore becomes difficult to generalize the study findings. Nonetheless, the use of PRISMA guidelines for the article selection, coupled with the developed protocol by the authors which guided the various processes of data extraction significantly reduced the number of omitted articles and ultimately eliminated possible biases associated with the research validity. Also, a preliminary search conducted on Google Scholar, Citeseer, and SCOPUS identified, IEEE Explore, ACM Digital Library, ScienceDirect, and Wiley Online as the most appropriate databases containing many of the studies on image steganography techniques. The quality assessment metrics used for the data extraction further reduced biases. The keywords developed were also aimed at reducing biases. Ultimately, the objective was to ensure the articles selected were of good quality.

In conclusion, it was observed that GAN has become the most preferred image steganography technique, and machine learning-based algorithms such as DL, CNN, and GA are recently dominating image steganography research. The implication is that previously preferred traditional techniques such as LSB, DCT, and IWT algorithms are receiving less attention in image steganography. Future research could explore emerging technologies such as blockchain technology and artificial neural networks to strike an adequate balance between imperceptibility, robustness, and enhanced security for data protection on one hand, and high embedding payload capacity on the other hand.

## Supporting information

**S1 Appendix. Detailed list of reviewed studies (RS).**
(DOCX)

**S2 Appendix. PRISMA 2020 checklist for the study.**
(DOCX)

**S3 Appendix. Data sources retrieved from electronic databases.**
(XLSX)

## Author Contributions

**Conceptualization:** Richard Apau, Kwame Ofosuhene Peasah.

**Data curation:** Richard Apau, James Ben Hayfron-Acquah.

**Investigation:** Richard Apau.

**Methodology:** Richard Apau.

**Software:** Michael Asante.

**Supervision:** Michael Asante, Frimpong Twum, James Ben Hayfron-Acquah.

**Validation:** Michael Asante, Frimpong Twum, James Ben Hayfron-Acquah, Kwame Ofosuhene Peasah.

**Visualization:** Frimpong Twum.

**Writing – original draft:** Richard Apau.

**Writing – review & editing:** Michael Asante, Frimpong Twum, James Ben Hayfron-Acquah, Kwame Ofosuhene Peasah.

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
