## [Decision Letter · Decision Letter 0]

17 May 2024

PONE-D-24-14917Image Steganography Techniques for Resisting Statistical Steganalysis Attacks: A Systematic Literature ReviewPLOS ONE

Dear Dr. APAU,

Thank you for submitting your manuscript to PLOS ONE. After careful consideration, we feel that it has merit but does not fully meet PLOS ONE’s publication criteria as it currently stands. Therefore, we invite you to submit a revised version of the manuscript that addresses the points raised during the review process.

We look forward to receiving your revised manuscript.

Kind regards,

Serdar Solak

Academic Editor

PLOS ONE

Journal Requirements:

2. Please remove your figures from within your manuscript file, leaving only the individual TIFF/EPS image files, uploaded separately. These will be automatically included in the reviewers’ PDF.

**Additional Editor Comments:**

The article should be subjected to a comprehensive review. Given that the article is a review study, it would be beneficial to include additional studies. The article should be updated in light of the suggestions provided by the referees. Some sections that are not essential should be shortened.

Reviewers' comments:

Reviewer's Responses to Questions

**Comments to the Author**

1. Is the manuscript technically sound, and do the data support the conclusions?

Reviewer #1: No

Reviewer #2: Yes

Reviewer #3: Yes

Reviewer #4: Yes

2. Has the statistical analysis been performed appropriately and rigorously? 

Reviewer #1: No

Reviewer #2: Yes

Reviewer #3: Yes

Reviewer #4: Yes

3. Have the authors made all data underlying the findings in their manuscript fully available?

Reviewer #1: No

Reviewer #2: Yes

Reviewer #3: Yes

Reviewer #4: Yes

4. Is the manuscript presented in an intelligible fashion and written in standard English?

Reviewer #1: Yes

Reviewer #2: Yes

Reviewer #3: Yes

Reviewer #4: Yes

5. Review Comments to the Author

Reviewer #1: The importance of the article and its contribution to the literature are not reflected in the abstract. The abstract should include the context or background information for your research; the general topic under study; the specific topic of your research; why is it important to address these questions; the significance or implications of your findings or arguments. It must also contain numeric values. Please highlight your contribution. Reorganize the abstract to conclude:

(a) The overall purpose of the study and the research problems you investigated.

(b) The basic design of the study.

(c) Major findings or trends found as a result of the study.

(d) A brief summary of your interpretations and conclusions.

Add more recent reference to enhance introduction section. Discuss the state-of-art techniques with their merits and issues. The literature should be developed and, if possible, presented in more papers published in last years. Discuss the research gaps and relate how the proposed work has improved them. In the introduction section, include spatial domain methods in your article. Also mention the difference between frequency and spatial domain methods For examples, “High embedding capacity data hiding technique based on EMSD and LSB substitution algorithms”, “Image steganography based on LSB substitution and encryption method: adaptive LSB+ 3”, “A New Dual Image Based Reversible Data Hiding Method Using Most Significant Bits and Center Shifting Technique” and “Data Hiding Based on Frequency Domain Image Steganography”

You should submit more experimental study results for your work. You should also provide comparisons with similar studies.

What solution you propose to make the system more robust. What is your difference from similar studies?

Prepare and explain your numerical examples more clearly for data hiding and data extraction processes.

In the study, it is necessary to present complexity analysis and security analysis. For security tests, histogram analysis and PDH can be given.

You must also provide numeric values. Rewrite the conclusion with following comments:(a) Highlight your analysis and reflect only the important points for the whole paper. (b) Mention the implication in the last of this section. Please, carefully review the manuscript to resolve these issues. (c) This section should be supported with numerical values.

There are grammatical errors in the article.

Reviewer #2: 1-The abstract contains more information than it should, for example, the sites that have been approved for papers and future work for the research presented. It is best to approach this matter in summary.

2- Please provide the researcher's contributions in point form

3- The equations listed are not referenced to the references on which they were based. In addition, it would have been better to include performance evaluation metrics in a table showing each measure and its equation

4- Is it possible to include a figure showing the superiority of the methods mentioned in Table 9 over other methods listed in the same table to demonstrate the efficiency of each method through the approved standards?

5- Is it possible to arrange the references in Table 9 according to the year of publication for ease of follow-up?

6- In Table 10, please add a field that shows the ratio of the number of images used to the maximum number of images used and discuss this matter due to its importance. I find it best to separate the table into two tables, the first for grayscale images and the other for color images for easy follow-up.

7-Rewriting in point form of Section 4.6 Future scope and research directions for image steganography

8- The research still needs to include other research that uses biometric techniques to hide images, such as voice or facial features. Please add such research to existing references

Reviewer #3: This paper presents an interesting image steganography overview. It covered systematic literature review of many stego techniques capable of resisting steganalysis attacks sampled from ACM Digital Library, IEEE Explore, Science Direct, and Wiley. The systematic review and Meta-Analyses have been synthesized and analysed using quantitative and qualitative methods. The works security and robustness are having significance, but the overall research presentation needs to be slightly improved covering some current aspects in order to be ready as publication, as noted within the following points that have to be fulfilled:

1- Give more elaboration on the real need for utilizing this Generative Adversarial Networks stego approaches. What is wrong in the normal other related stego methods requiring this kind of complex research. Try to support your explanation via real-life examples.

2- The study is recommended to consider the following image steganalysis researches within its coverage:

== "Integrating machine learning and features extraction for practical reliable color images steganalysis classification", Soft Computing 27(19):13877-13888 (2023)

== "Towards improving the performance of blind image steganalyzer using third-order SPAM features and ensemble classifier", Journal of Information Security and Applications 76:10354, Elsevier (2023)

== “Is blind image steganalysis practical using feature-based classification?”. Multimedia Tools and Applications (MTAP) 83(2): 4579–4612 (2024)

3- The image security review presented is very promising showing its high capacity trade-off to security and robustness improvement as challenging processes. Try to benefit from the image stego secrecy researches below connecting to the capacity vs secrecy of stego hiding attempts provided:

== "Efficient Reversible Data Hiding Multimedia Technique Based on Smart Image Interpolation", Multimedia Tools and Applications (MTAP) 79(39):30087-30109 (2020)

== "Novel Embedding Secrecy within Images Utilizing an Improved Interpolation-Based Reversible Data Hiding Scheme", Journal of King Saud University - Computer and Information Sciences, 34(5):2017-2030 (2022)

==  "Efficient Implementation of Multi-image Secret Hiding Based on LSB and DWT Steganography Comparisons", Arabian Journal for Science and Engineering (AJSE) 45:2631–2644 (2020)

== "High performance image steganography integrating IWT and Hamming code within secret sharing", IET Image Processing, 18(1): 129-139 (2024)

== “Dynamic smart random preference for higher medical image confidentiality”, Journal of Engineering Research (JER) 11(3A): 100-111 (2023)

== "Vibrant Color Image Steganography using Channel Differences and Secret Data Distribution", Kuwait Journal of Science and Engineering (KJSE) 38(1B):127-142 (2011)

==  "Pixel Indicator Technique for RGB Image Steganography", Journal of Emerging Technologies in Web Intelligence (JETWI) 2(1):56-64 (2010)

== “Improving grayscale steganography to protect personal information disclosure within hotel services”, Multimedia Tools and Applications (MTAP) 81(21): 30663–30683 (2022)

== "Improving data hiding within colour images using hue component of HSV colour space", CAAI Transactions on Intelligence Technology, IET (IEE) - Wiley, 7(1): 56–68 (2022)

==  "Efficient Image Reversible Data Hiding Technique Based on Interpolation Optimization", Arabian Journal for Science and Engineering (AJSE) 46(9):8441–8456 (2021)

== "Trustworthy image security via involving binary and chaotic gravitational searching within PRNG selections", International Journal of Computer Science and Network Security (IJCSNS) 20(12):167-176 (2020)

4- Combining cryptography and steganography is used a lot in literature as very promising philosophy to stand the privacy system against many security attacks. Although this combination is noted as extra payload in this direction, briefly link your review study work to the complexity combinations within the following studies:

== "Enhancing Medical Data Security via Combining Elliptic Curve Cryptography with 1-LSB and 2-LSB Image Steganography", International Journal of Computer Science and Network Security (IJCSNS) 20(12):232-241 (2020)

== "Protecting Medical Records against Cybercrimes within Hajj Period by 3-layer Security", Recent Trends in Information Technology and Its Application 2(3):1–21 (2019)

== "Enhancing Medical Data Security via Combining Elliptic Curve Cryptography and Image Steganography", International Journal of Computer Science and Network Security (IJCSNS) 20(8):1-8 (2020)

== "Securing Matrix Counting-Based Secret-Sharing Involving Crypto Steganography", Journal of King Saud University - Computer and Information Sciences, ISSN:1319-1578, 34(9): 6909–6924 (2022)

== "Watermarking Images via Counting-Based Secret Sharing for Lightweight Semi-Complete Authentication",  International Journal of Information Security and Privacy (IJISP) 16(1): 1-18 (2022)

== "Increasing Participants Using Counting-Based Secret Sharing via Involving Matrices and Practical Steganography", Arabian Journal for Science and Engineering (AJSE), 47(2): 2455–2477 (2022)

== "Refining image steganography distribution for higher security multimedia counting-based secret-sharing", Multimedia Tools and Applications (MTAP) 80:1143–1173 (2021)

== "Enhancing PC Data Security via Combining RSA Cryptography and Video Based Steganography", Journal of Information Security and Cybercrimes Research (JISCR) 1(1):5-13 (2018)

== "Compression Multi-Level Crypto Stego Security of Texts Utilizing Colored Email Forwarding", Journal of Computer Science & Computational Mathematics (JCSCM) 8(3):33-42 (2018)

== “3-Layer PC Text Security via Combining Compression, AES Cryptography 2LSB Image Steganography”, Journal of Research in Engineering and Applied Sciences (JREAS) 3(4):118-124 (2018)

== "Enhancing Speed of SIMON: A Light-Weight-Cryptographic Algorithm for IoT Applications", Multimedia Tools and Applications (MTAP) 78:32633–32657 (2019)

Reviewer #4: Title:

Image Steganography Techniques for Resisting Statistical Steganalysis Attacks: A Systematic Literature Review

Comments:

This article is a review article, so there is nothing new approach related to steganography techniques to be discussed. This article is well-written technically. And, although there have been many reviews of articles on image steganography, this article can still be considered because there is a special appreciation for the statistical approaches that are focused on this article.

However, there are important points that can be taken into consideration to improve the content of this article. In RQ2, the author tries to explore steganography techniques and methods that are currently widely used to deal with attacks. It would be good if the discussion of RQ2 also discussed the issues that are generally raised in research on steganography. So, it will be seen that these techniques/methods are proposed to overcome certain problems.

The connection between the problem/issue and the technique/method becomes clearer before discussing the advantages/weaknesses of a method. In short, if a 'problem/issue' column could be inserted before the 'technique/method' column, it seems that Table 7 would be more useful.

6. PLOS authors have the option to publish the peer review history of their article (what does this mean?). If published, this will include your full peer review and any attached files.

Reviewer #1: No

Reviewer #2: No

Reviewer #3: No

Reviewer #4: No

---

## [Author Response · Author response to Decision Letter 0]

2 Jul 2024

Editor Comments:

Comments1: The article should be subjected to a comprehensive review. Given that the article is a review study, it would be beneficial to include additional studies. The article should be updated in light of the suggestions provided by the referees. Some sections that are not essential should be shortened.

Response: We have comprehensively reviewed the article as per the suggestions. We have also included some additional studies, in line with suggestions of reviewers and the judgement of the editor, even though we believed our work contained enough recent studies on the subject. Some sections which we considered not essential have been reduced and some figures taken out. This can be seen in the tracked change revised manuscript. A total of 20 additional references were included as per reviewers’ recommendations. 

We, however, want to bring to the attention of the editor, on what we term as Citation Forum Shopping as a consequence of the suggested articles to us by the reviewers. We respect the need for anonymity in academic peer reviewing. However, based on reference suggestions by some of the reviewers, which they termed recent publication, which are not anyway, Solak and Gutub could easily be identified as potential reviewers of our work. While we respect their expertise in the field, there are many more diverse researchers in this area, and we were surprised that all articles suggested had one particular name running through it. This may not be ethical, and we feel we must draw the attention of the editor to avoid such practices from reviewers in the Future. We do respect and value the high reputation of PLOS ONE in academic publication. Of course, most of the papers were not cited by us, as we did not consider most of them relevant to our work based on our own quality criteria. Those that we feel align to the objectives of our review were considered, though a handful as some of the suggested articles bear no semblance of our work. In the future, article suggestions from reviewers could be checked to avoid a situation where reviewers may want to take advantage of peer reviewing to force their papers on authors for purposes of citation counts. 

See Paper Suggestions by Reviewer 1.

1. Solak S, Tezcan G. A New Dual Image Based Reversible Data Hiding Method Using Most Significant Bits and Center Shifting Technique. Applied Sciences. 2022 Oct 28;12(21):10933.

2. Abdirashid AM, Solak S, Sahu AK. Data Hiding Based on Frequency Domain Image Steganography. Avrupa Bilim ve Teknoloji Dergisi. 2022(42):71-6.

3. Solak S. High embedding capacity data hiding technique based on EMSD and LSB substitution algorithms. IEEE Access. 2020 Sep 10;8:166513-24.

4. Solak S, Altınışık U. Image steganography based on LSB substitution and encryption method: adaptive LSB+ 3. Journal of Electronic Imaging. 2019 Jul 1;28(4):043025-.

See Paper Suggestions by Reviewer 3

1. Aljarf A, Zamzami H, Gutub A. Integrating machine learning and features extraction for practical reliable color images steganalysis classification. Soft Computing. 2023 Oct;27(19):13877-88.

2. Hemalatha J, Sekar M, Kumar C, Gutub A, Sahu AK. Towards improving the performance of blind image steganalyzer using third-order SPAM features and ensemble classifier. Journal of Information Security and Applications. 2023 Aug 1;76:103541.

3. Aljarf A, Zamzami H, Gutub A. Is blind image steganalysis practical using feature-based classification?. Multimedia Tools and Applications. 2024 Jan;83(2):4579-612.

4. Hassan FS, Gutub A. Efficient reversible data hiding multimedia technique based on smart image interpolation. Multimedia Tools and Applications. 2020 Oct;79(39):30087-109.

5. Hassan FS, Gutub A. Novel embedding secrecy within images utilizing an improved interpolation-based reversible data hiding scheme. Journal of King Saud University-Computer and Information Sciences. 2022 May 1;34(5):2017-30.

6. Gutub A, Al-Shaarani F. Efficient implementation of multi-image secret hiding based on LSB and DWT steganography comparisons. Arabian Journal for Science and Engineering. 2020 Apr;45(4):2631-44.

7. Saeidi Z, Yazdi A, Mashhadi S, Hadian M, Gutub A. High performance image steganography integrating IWT and Hamming code within secret sharing. IET Image Processing. 2024 Jan;18(1):129-39.

8. Gutub A. Dynamic smart random preference for higher medical image confidentiality. Journal of Engineering Research. 2023;11(3).

9. Parvez MT, Gutub AA. Vibrant color image steganography using channel differences and secret data distribution. Kuwait J Sci Eng. 2011 Jun 1;38(1B):127-42.

10. Gutub AA. Pixel indicator technique for RGB image steganography. Journal of emerging technologies in web intelligence. 2010 Feb;2(1):56-64.

11. Sahu AK, Gutub A. Improving grayscale steganography to protect personal information disclosure within hotel services. Multimedia Tools and Applications. 2022 Sep;81(21):30663-83.

12. Hassan FS, Gutub A. Improving data hiding within colour images using hue component of HSV colour space. CAAI Transactions on Intelligence Technology. 2022 Mar;7(1):56-68.

13. Hassan FS, Gutub A. Efficient image reversible data hiding technique based on interpolation optimization. Arabian Journal for Science and Engineering. 2021 Sep;46(9):8441-56.

14. Al-Roithy B, Gutub A. Trustworthy image security via involving binary and chaotic gravitational searching within PRNG selections. Int. J. Comput. Sci. Netw. Secur. 2020 Dec;20(12):167-76.

15. Hureib ES, Gutub AA. Enhancing medical data security via combining elliptic curve cryptography and image steganography. Int. J. Comput. Sci. Netw. Secur.(IJCSNS). 2020 Aug;20(8):1-8.

16. Samkari H, Gutub A. Protecting medical records against cybercrimes within Hajj period by 3-layer security. Recent Trends Inf Technol Appl. 2019 Nov 15;2(3):1-21.

17. Hureib ES, Gutub AA. Enhancing medical data security via combining elliptic curve cryptography with 1-LSB and 2-LSB image steganography. International J Comp Sci Network Security (IJCSNS). 2020 Dec;20(12):232-41.

18. Samkari H, Gutub A. Protecting medical records against cybercrimes within Hajj period by 3-layer security. Recent Trends Inf Technol Appl. 2019 Nov 15;2(3):1-21.

19. Al-Shaarani F, Gutub A. Securing matrix counting-based secret-sharing involving crypto steganography. Journal of King Saud University-Computer and Information Sciences. 2022 Oct 1;34(9):6909-24.

20. Gutub A. Watermarking images via counting-based secret sharing for lightweight semi-complete authentication. International Journal of Information Security and Privacy (IJISP). 2022 Jan 1;16(1):1-8.

21. Al-Shaarani F, Gutub A. Increasing participants using counting-based secret sharing via involving matrices and practical steganography. Arabian Journal for Science and Engineering. 2022 Feb;47(2):2455-77.

22. AlKhodaidi T, Gutub A. Refining image steganography distribution for higher security multimedia counting-based secret-sharing. Multimedia Tools and Applications. 2021 Jan;80:1143-73.

23. Al-Juaid NA, Gutub AA, Khan EA. Enhancing PC data security via combining RSA cryptography and video based steganography. Journal of Information Security and Cybercrimes Research. 2018 Apr 17;1(1):5-13.

24. Alsaidi A, Al-lehaibi K, Alzahrani H, AlGhamdi M, Gutub A. Compression multi-level crypto stego security of texts utilizing colored email forwarding. Journal of Computer Science & Computational Mathematics (JCSCM). 2018 Sep;8(3):33-42.

25. Alassaf N, Gutub A, Parah SA, Al Ghamdi M. Enhancing speed of SIMON: A light-weight-cryptographic algorithm for IoT applications. Multimedia Tools and Applications. 2019 Dec;78:32633-57.

Reviewer #1: 

Given that the comments were not numbered, we have decided to group the comments for easy response. 

Comment 1: The importance of the article and its contribution to the literature are not reflected in the abstract. The abstract should include the context or background information for your research; the general topic under study; the specific topic of your research; why is it important to address these questions; the significance or implications of your findings or arguments. It must also contain numeric values. Please highlight your contribution. Reorganize the abstract to conclude:

(a) The overall purpose of the study and the research problems you investigated.

(b) The basic design of the study.

(c) Major findings or trends found as a result of the study.

(d) A brief summary of your interpretations and conclusions.

Response: We believe this comment does not reflect the information contained in our paper. The abstract has all relevant information, including the background, summary of interpretations and conclusions drawn from our review. We must indicate that, our abstract is in line with the PRISMA 2020 check list for systematic review, which is a mandatory requirement by PLOS ONE. We tailored our abstract to fulfil PLOS ONE requirements in the checklist. 

Comments 2: Add more recent reference to enhance introduction section. Discuss the state-of-art techniques with their merits and issues. The literature should be developed and, if possible, presented in more papers published in last years. Discuss the research gaps and relate how the proposed work has improved them. In the introduction section, include spatial domain methods in your article. Also mention the difference between frequency and spatial domain methods For examples, “High embedding capacity data hiding technique based on EMSD and LSB substitution algorithms”, “Image steganography based on LSB substitution and encryption method: adaptive LSB+ 3”, “A New Dual Image Based Reversible Data Hiding Method Using Most Significant Bits and Center Shifting Technique” and “Data Hiding Based on Frequency Domain Image Steganography”

Response: The issues raised have been well discussed in our background literature. Spatial domain, frequency domain and Adaptive domain have all been adequately addressed. Please refer section 2 on background literature. Nonetheless, we have improved our introduction and included two of the suggested references. 

Comments 3: You should submit more experimental study results for your work. You should also provide comparisons with similar studies.

What solution you propose to make the system more robust. What is your difference from similar studies?Prepare and explain your numerical examples more clearly for data hiding and data extraction processes.In the study, it is necessary to present complexity analysis and security analysis. For security tests, histogram analysis and PDH can be given.You must also provide numeric values. Rewrite the conclusion with following comments:(a) Highlight your analysis and reflect only the important points for the whole paper. (b) Mention the implication in the last of this section. Please, carefully review the manuscript to resolve these issues. (c) This section should be supported with numerical values. There are grammatical errors in the article.

Response: We want to believe that Reviewer 1 may have misconstrued our paper to be an original image steganography experiment work. Our work is a systematic review, and we did not conduct primary experiment. Most of the things being requested are things we do for primary steganography applications. We cannot do Complexity and Steganalysis security tests. These can only be done by original authors of the papers we used. Where the authors performed such tests, we reported, where they did not, we cannot do anything about them. 

Reviewer #2: 

Comment 1: The abstract contains more information than it should, for example, the sites that have been approved for papers and future work for the research presented. It is best to approach this matter in summary.

Response: As a systematic review article, we have to strictly follow the PLOS ONE 2020 PRISMA checklist for systematic review abstract. It is the reason why, it appears more information are included than necessary. This is a framework requirement. We have however made some grammatical corrections in the abstract. 

Comment 2: Please provide the researcher's contributions in point form

Response: This has been done in the revised manuscript. Contribution has been written in point form as suggested by the reviewer. 

Comment 3: The equations listed are not referenced to the references on which they were based. In addition, it would have been better to include performance evaluation metrics in a table showing each measure and its equation

Response: All equations have been referenced as suggested, although the source of each equations were already mentioned in the paper. We tried to put the equations in a table form but the presentation was not good for the flow of the work. We therefore left it as it was. 

Comment 4: Is it possible to include a figure showing the superiority of the methods mentioned in Table 9 over other methods listed in the same table to demonstrate the efficiency of each method through the approved standards?

Response: We have included Figure 7 to comply with this suggested comment

Comment 5: Is it possible to arrange the references in Table 9 according to the year of publication for ease of follow-up?

Response: We have implemented this as suggested. Table 7,8, and 9 have been arranged in that order.

Comment 6: In Table 10, please add a field that shows the ratio of the number of images used to the maximum number of images used and discuss this matter due to its importance. I find it best to separate the table into two tables, the first for grayscale images and the other for color images for easy follow-up.

Response: We have separated table 10 into two for grayscale and color images as suggested. We now have table 10 and table 11. The ratio of the number of images used to the maximum number of images used is called the capacity, which is already part of our performance evaluation metrics. These results are reported in table 7,8, and 9 in the last column. Capacity is measured in many ways, some use ratio, others used capacity as a percentage whereas some studies used bit rate (BPP). We have reported depending on what the authors used. Please refer to the tables mentioned. 

Comment 7: Rewriting in point form of Section 4.6 Future scope and research directions for image steganography

Response: We have done this as suggested. Please refer to the revised manuscript.

Comment 8: The research still needs to include other research that uses biometric techniques to hide images, such as voice or facial features. Please add such research to existing references

Response: We have included these techniques in our explanations as suggested. See section 2 for background literature and references 59-63. 

Reviewer #3: 

Comment 1- Give more elaboration on the real need for utilizing this Generative Adversarial Networks stego approaches. What is wrong in the normal other related stego methods requiring this kind of complex research. Try to support your explanation via real-life examples.

Response: These comments have been resolved in the revised manuscript. Explanations have been provided as suggested. Please refer to section 5, subsection 5.2 for the revision on the comments. 

Comment 2- The study is recommended to consider the following image steganalysis researches within its coverage:

== "Integrating machine learning and features extraction for practical reliable color images steganalysis classification", Soft Computing 27(19):13877-13888 (2023)

== "Towards improving the performance of blind image steganalyzer using third-order SPAM features and ensemble classifier", Journal of Information Security and Applications 76:10354, Elsevier (2023)

== “Is blind image steganalysis practical using feature-based classification?”. Multimedia Tools and Applications (MTAP) 83(2): 4579–4612 (2024)

Response: The above articles have been referenced as we consider relevant to our work particularly on the background literature. Check reference 74-76.

Comment 3: The image security review presented is very promising showing its high capacity trade-off to security and robustness improvement as challenging processes. Try to benefit from the image stego secrecy researches below connecting to the capacity 

---

## [Decision Letter · Decision Letter 1]

16 Jul 2024

PONE-D-24-14917R1Image Steganography Techniques for Resisting Statistical Steganalysis Attacks: A Systematic Literature ReviewPLOS ONE

Dear Dr. APAU,

Thank you for submitting your manuscript to PLOS ONE. After careful consideration, we feel that it has merit but does not fully meet PLOS ONE’s publication criteria as it currently stands. Therefore, we invite you to submit a revised version of the manuscript that addresses the points raised during the review process.

We look forward to receiving your revised manuscript.

Kind regards,

Academic Editor

PLOS ONE

Journal Requirements:

Additional Editor Comments:

Authors should reflect the importance of the article in the abstract. The abstract is too long, please review and shorten it. Although 125 references are also mentioned in the abstract, there are 147 items in the references section.

Use more academic language, have the article checked for grammar.

Check your section numbers.

Review your tables, figures and equations again.

You should increase the figure quality.

Check your fonts and size.

Reviewers' comments:

Reviewer's Responses to Questions

**Comments to the Author**

1. If the authors have adequately addressed your comments raised in a previous round of review and you feel that this manuscript is now acceptable for publication, you may indicate that here to bypass the “Comments to the Author” section, enter your conflict of interest statement in the “Confidential to Editor” section, and submit your "Accept" recommendation.

Reviewer #2: All comments have been addressed

Reviewer #3: All comments have been addressed

Reviewer #4: All comments have been addressed

2. Is the manuscript technically sound, and do the data support the conclusions?

Reviewer #2: Yes

Reviewer #3: Yes

Reviewer #4: Yes

3. Has the statistical analysis been performed appropriately and rigorously? 

Reviewer #2: Yes

Reviewer #3: Yes

Reviewer #4: Yes

4. Have the authors made all data underlying the findings in their manuscript fully available?

Reviewer #2: Yes

Reviewer #3: Yes

Reviewer #4: Yes

5. Is the manuscript presented in an intelligible fashion and written in standard English?

Reviewer #2: Yes

Reviewer #3: Yes

Reviewer #4: Yes

6. Review Comments to the Author

Reviewer #2: (No Response)

Reviewer #3: Thanks for the hard work. The revision is performed perfect. All comments are addressed in satisfying condition.

Reviewer #4: Authors have revised and completed the manuscript according to my review from the previous round, especially to emphasize the RQ2 in their surveys.

The discussion about RQ2 is now more complete with an exploration of methods along with the relevant problems. Additional information in Tables 7, 8, and 9 as well as the given brief explanation can enrich the reader's knowledge, especially on research topics in image hiding and steganography fields.

7. PLOS authors have the option to publish the peer review history of their article (what does this mean?). If published, this will include your full peer review and any attached files.

Reviewer #2: No

Reviewer #3: No

Reviewer #4: No

---

## [Author Response · Author response to Decision Letter 1]

24 Jul 2024

Editor Comments:

Comment 1: Authors should reflect the importance of the article in the abstract. 

Response: We have included in the abstract the main aim of the article and its importance to image steganography researchers and data protection practitioners. 

Comment 2: The abstract is too long, please review and shorten it. 

Response: This has been addressed. We have reviewed and shortened the abstract from the initial 300 words to 250 words. 

Comment 3: Although 125 references are also mentioned in the abstract, there are 147 items in the references section.

Response: There is a difference between the 125 articles referenced in the abstract and the reference list generated at the end of the article. The 125 mentioned in the abstract is the total number of articles used to perform the systematic literature review. All those articles have been added at the end as APPENDIX I (RS1-RS125). This is separate from works cited in the paper which is the reference list of 148. 

Comment 4: Use more academic language, have the article checked for grammar.

Response: we have performed academic language review and corrected where necessary. We have also used software including Grammarly to correct identified grammatical errors. 

Comment 5: Check your section numbers.

Response: Ce have checked our section numbers. They are all sequential and correctly numbered. 

Comment 6: Review your tables, figures and equations again.

Response: Table, figures and equations have been reviewed again to ensure they are correctly labelled and titled. 

Comment 7: You should increase the figure quality.

Response: Figures whose quality was deemed low have been regenerated to ensure high resolution. 

Comment 8: Check your fonts and size.

Response: Font size throughout the article have been set to times new roman 11 as per PLOS ONE specifications, except in tables where a lower font size is chosen to fit the words. 

Comments by REVIEWERS:

We noticed that all reviewers accepted the revised manuscript as fully meeting their review requirements without further review. All reviewers appreciated the revised manuscript.

---

## [Editor Report · Decision Letter 2]

26 Jul 2024

Image Steganography Techniques for Resisting Statistical Steganalysis Attacks: A Systematic Literature Review

PONE-D-24-14917R2

Dear Dr. APAU,

We’re pleased to inform you that your manuscript has been judged scientifically suitable for publication and will be formally accepted for publication once it meets all outstanding technical requirements.

Kind regards,

Academic Editor

PLOS ONE

Additional Editor Comments (optional):

After carefully reviewing the revised manuscript titled "Image Steganography Techniques for Resisting Statistical Steganalysis Attacks: A Systematic Literature Review" and considering the authors' responses to the reviewers' and editor's suggestions, I find that the authors have made the necessary revisions to address all concerns raised.

The manuscript is now appropriate for acceptance.

---

## [Editor Report · Acceptance letter]

1 Aug 2024

PONE-D-24-14917R2 

PLOS ONE

Dear Dr. APAU, 

I'm pleased to inform you that your manuscript has been deemed suitable for publication in PLOS ONE. Congratulations! Your manuscript is now being handed over to our production team.

Kind regards, 

on behalf of

Assoc. Prof. Serdar Solak 

Academic Editor

PLOS ONE